

# Large fluctuations of the KPZ equation in a half-space

**Alexandre Krajenbrink[1]★ and Pierre Le Doussal[1]**

**1** Laboratoire de Physique Théorique de l'Ecole Normale Supérieure
PSL University, CNRS, Sorbonne Universités
24 rue Lhomond, 75231 Paris Cedex 05, France

★ krajenbrink@ens.fr

## Abstract

We investigate the short-time regime of the KPZ equation in $1+1$ dimensions and develop a unifying method to obtain the height distribution in this regime, valid whenever an exact solution exists in the form of a Fredholm Pfaffian or determinant. These include the droplet and stationary initial conditions in full space, previously obtained by a different method. The novel results concern the droplet initial condition in a half space for several Neumann boundary conditions: hard wall, symmetric, and critical. In all cases, the height probability distribution takes the large deviation form $P(H, t) \sim \exp(-\Phi(H)/\sqrt{t})$ for small time. We obtain the rate function $\Phi(H)$ analytically for the above cases. It has a Gaussian form in the center with asymmetric tails, $|H|^{5/2}$ on the negative side, and $H^{3/2}$ on the positive side. The amplitude of the left tail for the half-space is found to be half the one of the full space. As in the full space case, we find that these left tails remain valid at all times. In addition, we present here (i) a new Fredholm Pfaffian formula for the solution of the hard wall boundary condition and (ii) two Fredholm determinant representations for the solutions of the hard wall and the symmetric boundary respectively.

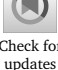
# 1   Introduction

Many recent works study the continuum KPZ equation in one dimension [1–7] which describes the stochastic growth of an interface parameterized by a height field $h(x,t)$ at point $x$ and time $t$ as

$$\partial_t h(x,t) = \nu \partial_x^2 h(x,t) + \frac{\lambda_0}{2} (\partial_x h(x,t))^2 + \sqrt{D}\, \xi(x,t)\,, \tag{1}$$

starting from a given initial condition $h(x,t=0)$. Here $\xi(x,t)$ is a centered Gaussian white noise with $\mathbb{E}\left[\xi(x,t)\xi(x',t')\right] = \delta(x-x')\delta(t-t')$, and we use from now on units of space, time and heights[1,2] such that $\lambda_0 = D = 2$ and $\nu = 1$ . Exact solutions have been found for several initial conditions, notably flat, droplet and stationary [8–16], and, remarkably, can be expressed using Fredholm determinants or Pfaffians. The typical behavior of the KPZ height fluctuations has been obtained from them, and related to the so-called Tracy Widom distributions in the large time limit (i.e. the distributions of the largest eigenvalues of standard Gaussian random matrix ensembles).

Recently, the *large deviations* away from the typical behavior have been studied. Unlike diffusive interacting particle systems for which powerful methods [17,18] were developed, systems in the KPZ class have required to develop new theoretical methods. A number of results have been obtained for the short time regime $t \ll 1$. They all agree that the probability density function (PDF), $P(H,t)$, of the properly shifted height at one space point $x=0$, denoted $H$, takes the large deviation form

$$\log P(H,t) \simeq_{t \ll 1} -\frac{\Phi(H)}{\sqrt{t}}, \tag{2}$$

where $\Phi(H)$ is the short time large deviation rate function, which depends on the initial condition. Two independent methods have been developed to show (2) and obtain properties of the rate function. The first method is the weak noise theory (WNT), pioneered in Ref. [19–21], which allows to obtain $\Phi(H)$ (i) for any $H$, from a numerical solution of saddle point differential equations (ii) analytically in the limits of large $|H|$ (and small $|H|$) for a variety of initial conditions [22–27]. The second method uses the exact solutions mentioned above and lead to an exact formula for $\Phi(H)$ for arbitrary $H$. It has been achieved for the droplet initial condition (IC) [28] (with an impressive confirmation from high precision numerics [29]) and for the stationary IC [30]. Remarkably, it has been recently shown that the exact formula for the flat IC is also contained in Ref. [30], up to a proper rescaling (i.e. one has $\Phi_{\text{flat}}(H) = 2^{-3/2}\Phi_{\text{stat}}(2H)$ choosing the analytic branch contained in Eqs. (28,29,30,31) of Ref. [30] with the label *analytic*). All of these results however concern the KPZ equation in the full space.

Here, we consider the KPZ equation in a half-space, where Eq. (1) is considered for $x \in \mathbb{R}^+$ along with the Neumann boundary condition (b.c.)

$$\forall t > 0, \quad \partial_x h(x,t)\,|_{x=0} = A, \tag{3}$$

where $A$ is a real parameter which describes the interaction with the boundary (a wall at $x=0$). This problem was considered in a pioneering paper by Kardar [31] in the equivalent representation in terms of a directed polymer near a wall. An unbinding transition to the wall

---

[1]Note that Refs. [22–25] use a sign of $\lambda_0$ opposite to ours (we use the same notations as in Refs. [28,30]). Equivalently, the variable $H$ is opposite to ours, which exchanges right and left tails.

[2]This is equivalent to use everywhere the following units of space, time and heights :
$x^* = (2\nu)^3/(D\lambda_0^2), \quad t^* = 2(2\nu)^5/(D^2\lambda_0^4), \quad h^* = \frac{2\nu}{\lambda_0}$

was predicted for $A = -1/2$ (and later observed in numerical simulations [32]). Here and below we restrict to the droplet IC

$$h(x, t = 0) = -\frac{|x - \varepsilon|}{\delta} - \log \delta, \tag{4}$$

with $\delta \ll 1$ and where $\varepsilon = 0^+$ is introduced to regularize the solution when it is not properly defined at $x = 0$, as it is the case for $A = +\infty$.

Exact solutions of this half-space problem, for the height at the origin at all times $t$, have been obtained in three cases [33–35] and can be expressed in terms of Fredholm Pfaffian. For $A = +\infty$, which corresponds to an (absorbing) hard wall in terms of the directed polymer, the PDF of the height converges at large time, in the typical regime $H \sim t^{1/3}$, to the Tracy-Widom (TW) distribution associated to the Gaussian Symplectic Ensemble (GSE) of random matrices [33]. For $A = 0$, which corresponds to a reflecting wall (a.k.a. the symmetric case), the large time limit of the PDF also corresponds to the TW-GSE distribution [34]. For $A = -\frac{1}{2}$, i.e. the critical case, the large time PDF in the typical regime is given by the Tracy Widom distribution associated to the GOE [35]. These exact solutions can be used to calculate the large deviations both in the short time and the large time regime. Concerning the large time, the tails of the PDF of the height in the typical regime $H \sim t^{1/3}$ are summarized in the following Table 1.

Table 1: Tails of the PDF of the centered height $H$ for large time $t \gg 1$ in the typical fluctuation regime $H \sim t^{1/3}$ in the various cases.

| ensemble | droplet IC | left tail $H \ll -t^{1/3}$ | right tail $H \gg t^{1/3}$ |
|----------|-----------|---------------------------|---------------------------|
| GUE | full-space | $e^{-\frac{1}{12}H^3/t}$ | $e^{-\frac{4}{3}H^{3/2}/t^{1/2}}$ |
| GOE | $A = -1/2$ | $e^{-\frac{1}{24}H^3/t}$ | $e^{-\frac{2}{3}H^{3/2}/t^{1/2}}$ |
| GSE | $A = 0, \infty$ | $e^{-\frac{1}{24}H^3/t}$ | $e^{-\frac{4}{3}H^{3/2}/t^{1/2}}$ |

As we discuss below these tails in typical regime are distinct from, but should match the large deviation tails discussed below. We now turn again the short time large deviations.

In this paper we use these exact solutions to establish (2) for the half-space problem and to calculate the large deviation rate function $\Phi(H)$ at short time for the above three cases. The method developed here generalizes the one introduced in [36] and applied there to the full space problem at large and short time respectively, and to the half-space critical case $A = -1/2$ in the large time regime. It is much simpler than the one used previously in [28,30]. It is based on the representation of a generating function of the KPZ field as an expectation value of a "Fermi factor" over a determinantal or Pfaffian point process. This expectation value can be expanded in cumulants, and its truncation to the first cumulant already yields the exact rate function $\Phi(H)$ at short time. Here we present this method in its most general formulation, so it can be applied readily to a variety of problems (including e.g. multicritical fermions, see [37]).

Our main results are listed in the following section and can be summarized as follows. The definition of the properly centered height $H$ (such that $\mathbb{E}[H] = 0$) in terms of $h(0, t)$ is given in Table 2 for each of the three cases. The rate function $\Phi(H)$ is determined from an auxiliary function $\Psi$, equivalently from (i) an implicit equation (11) (ii) a parametric system (12) and (13). The function $\Psi$, which is the large deviation rate function of the Fredholm determinant (or Pfaffian) itself, is determined by the density of the associated point process $\rho_\infty$ via equation (9). The density $\rho_\infty$ is given explicitly in the Table 2 for each of the three

cases (the factor $\chi$ is also given there). From these formula one can derive the explicit behavior around the center of the distribution, i.e. for small $|H|$, which is $\Phi(H) \simeq H^2/(2C_2)$ and the second cumulant $\mathbb{E}[H^2] \simeq C_2\sqrt{t}$ where $C_2$ is given in the Table 2. Higher orders in the power series expansion of $\Phi(H)$ and the higher cumulants of $H$ are also given in the Table 2.

The tails of the large deviation rate function are given in the Table 2. We find that the left tail exhibits the 5/2 exponent, $\Phi(H) \simeq_{H\to-\infty} \frac{2}{15\pi}|H|^{5/2}$, with an amplitude which is half of the amplitude of the full space solution with droplet initial conditions. For the right tail we find the usual 3/2 exponent, with two distinct cases for the amplitude. For $A = +\infty$ we find $\Phi(H) \simeq_{H\to+\infty} \frac{4}{3}H^{3/2}$ while for $A = 0, -1/2$ we find $\Phi(H) \simeq_{H\to+\infty} \frac{2}{3}H^{3/2}$ as indicated.

Furthermore, we find that for both values $A = -1/2$ and $A = 0$ the full rate function $\Phi(H)$ is equal to $\frac{1}{2}\Phi_{\text{full-space}}(H)$ (both for droplet initial conditions)[3]. In the two tails, this agrees with the result given in the previous paragraph, together with the formula given [23, 28] for the tails of the full space problem. Only the rate function for $A = +\infty$ is unrelated to the ones found in previous studies. Moreover, a perturbative expansion of the stochastic heat equation, equivalent to the KPZ equation, shows that the relevant parameter is $A\sqrt{t}$, hence at short time we expect that any finite $A$ has identical rate function. This is in agreement with the cases $A = 0$ and $A = -1/2$ explicitly solved here.

In addition to the interest in the large deviations of the KPZ equation at short time, recent works have studied the large deviations at large time $t \gg 1$ [36, 38–41]. The left tail was argued quite generally to take the form $\log P(H, t) \simeq_{t\gg1} -t^2\Phi_-(H/t)$ for large negative fluctuations $-H \sim t$. [38]. In recent works the explicit expression of $\Phi_-(z)$ for droplet initial conditions in the full space was obtained (i) using a WKB type approximation [39] on a non local Painleve type equation representation of the exact solution derived in [11] (ii) using Coulomb gas methods [41]. It exhibits a crossover between a cubic tail $\Phi_-(z) \simeq z^3/12$ (matching the Tracy Widom distribution [38]) and a 5/2 tail exponent $\Phi_-(z) \simeq \frac{4}{15\pi}z^{5/2}$. The latter can be readily obtained [36] using the method of truncation to the first cumulant described above, and is identical to the tail behavior at short time (a signature that the left tail remains identical at all times [41]). A similar result was obtained for the half-space with droplet initial condition and $A = -1/2$ in [41] with the result that $\Phi_-^{\text{half-space}}(z) = \frac{1}{2}\Phi_-^{\text{full-space}}(z)$ (see also [36]). In this paper we extend some of these results to the cases $A = 0$ and $A = +\infty$. We establish that in all cases $\log P(H, t) \simeq_{t\gg1} -\frac{2}{15\pi}H^{5/2}/t^{1/2}$ in the regime $-H/t \gg 1$. Further arguments leads us to conjecture that $\Phi_-^{\text{half-space}}(z) = \frac{1}{2}\Phi_-^{\text{full-space}}(z)$ for all $A > -1/2$. One can check that the small $z$ cubic behavior of these large deviation predictions matches perfectly the left tails of the typical regime see Table 1.

Finally one can ask how the tails evolve in time. From the results mentionned in the two previous paragraphs we see that the $|H|^{5/2}$ left tails have identical prefactor at short and large times for all cases. For the right $|H|^{3/2}$ tails we can compare the above results for short time and the tail behavior in the typical region in Table 1. The prefactors are identical for for $A = -1/2$ and $A = +\infty$ suggesting that the right tail is established at early times and does not change after that. However for $A = 0$ it has prefactor 2/3 for short time and 4/3 for large time, suggesting some evolution with time, or a more complex tail structure (which may be associated to $A = 0$ not being a critical fixed point)[4].

In the case of the hard wall, we have obtained a new useful representation of the exact solution at all times in terms of a Fredholm Pfaffian with a matrix valued kernel which we show is equivalent to the solution of [33] expressed in terms a Fredholm determinant with a scalar valued kernel. The interest of this representation is to provide a connection with a Pfaffian point process that converges at large time to the GSE. In addition, we have also

---

[3]Note for $A = 0$ a similar result was mentionned (without details) in [27] on the basis of WNT.

[4]We also rely on the conjectured exact solution in [34]

generalized this connection to a broader class of matrix valued kernels which should be useful to study further properties of determinantal point processes.

The outline of the paper is as follows. Two types of new results for the solutions of the KPZ equation in half space for the droplet IC are provided here. In Sections 3 and 4 we develop a unifying method to study the exact large deviation rate function $\Phi$ at short time, valid whenever a Pfaffian or determinantal representation of the exact solution is available. We apply this framework in Sections 5, 6 and 7 to the three cases $A = +\infty, -\frac{1}{2}, 0$. We provide in Section 5 a new kernel representation for the hard wall case $A = +\infty$ in terms of a Fredholm Pfaffian. In Section 8, we study the short time perturbation theory of the KPZ equation in half-space and argue that at short time there exists only two fixed points for the large deviation rate function $\Phi$ given by the $A = \infty$ case and the $A = 0$ case. Finally, in Section 9, following the approach of [36, 41], we show that the left tail, i.e. the left asymptotics of $\Phi$ remains valid at all times. In addition to some technical details present in Appendix A, C and D, we present in Appendix B our connection mentionned above between Fredholm Pfaffians with matrix valued kernels and Fredholm determinants with scalar valued kernels.

## 2 Presentation of the main results

We give in this section a summary of the new results of this paper. Firstly, we exhibit a new kernel representation for the hard wall case $A = +\infty$. Secondly, we present some general mathematical rules to obtain properties about the short time distribution of the KPZ solution. A visual summary is given in Table 2.

### 2.1 A new kernel for the hard wall

A new identity for the moment generating function of the Cole-Hopf solution of the KPZ equation for the hard wall $A = +\infty$ case is given for $z \geq 0$ as

$$\mathbb{E}_{\text{KPZ}}\left[\exp\left(-z e^{H_1}\right)\right] = 1 + \sum_{n_s=1}^{\infty} \frac{(-1)^{n_s}}{n_s!} \prod_{p=1}^{n_s} \int_{\mathbb{R}} \mathrm{d}r_p \frac{z}{z + e^{-t^{1/3}r_p}} \text{Pf}\left[K(r_i, r_j)\right]_{n_s \times n_s}, \qquad (5)$$

where $H_1 = h(\varepsilon, t) + \frac{t}{12} - 2\log\varepsilon$ for $\varepsilon \ll 1$, the expected value of the l.h.s of (5) is taken over the realization of the KPZ white noise, and $K$ is a $2 \times 2$ block matrix with elements

$$K_{11}(r, r') = \int_{C_v}\int_{C_w} \frac{\mathrm{d}v\mathrm{d}w}{(2i\pi)^2 \pi t^{\frac{1}{3}}} \frac{v-w}{v+w} \Gamma(2vt^{-\frac{1}{3}})\Gamma(2wt^{-\frac{1}{3}})\cos(\pi vt^{-\frac{1}{3}})\cos(\pi wt^{-\frac{1}{3}})e^{-rv-r'w+\frac{v^3+w^3}{3}},$$

$$K_{22}(r, r') = \int_{C_v}\int_{C_w} \frac{\mathrm{d}v\mathrm{d}w}{(2i\pi)^2 \pi t^{\frac{1}{3}}} \frac{v-w}{v+w} \Gamma(2vt^{-\frac{1}{3}})\Gamma(2wt^{-\frac{1}{3}})\sin(\pi vt^{-\frac{1}{3}})\sin(\pi wt^{-\frac{1}{3}})e^{-rv-r'w+\frac{v^3+w^3}{3}},$$

$$K_{12}(r, r') = \int_{C_v}\int_{C_w} \frac{\mathrm{d}v\mathrm{d}w}{(2i\pi)^2 \pi t^{\frac{1}{3}}} \frac{v-w}{v+w} \Gamma(2vt^{-\frac{1}{3}})\Gamma(2wt^{-\frac{1}{3}})\cos(\pi vt^{-\frac{1}{3}})\sin(\pi wt^{-\frac{1}{3}})e^{-rv-r'w+\frac{v^3+w^3}{3}},$$

$$K_{21}(r, r') = -K_{12}(r', r).$$

$$(6)$$

The contours $C_v$ and $C_w$ must both pass at the right of $0$ because of the $\Gamma$ functions as $C_{v,w} = \frac{1}{2}a_{v,w} + i\mathbb{R}$ for $a_{v,w} \in ]0, t^{1/3}[$ and they must be such that $\text{Re}(v + w) > 0$ for the denominators to be well defined. We additionally have the Pfaffian point process identity for

$z \geq 0$

$$\mathbb{E}_{\text{KPZ}}\left[\exp\left(-ze^{H_1}\right)\right] = \mathbb{E}_K\left[\prod_{i=1}^{\infty}\frac{1}{1+ze^{t^{1/3}a_i}}\right], \tag{7}$$

where the r.h.s of (7) is an average over the Pfaffian point process with kernel $K$ that generates the set $\{a_i\}_{i\in\mathbb{N}}$. Note that it can also be written as a *Fredholm Pfaffian*, see Section 3.1. For the definition of a Pfaffian point process see [42–44]. For the definition and properties of Fredholm Pfaffians see Sec. 8 in [42], as well as e.g. Sec. 2.2. in [45], Appendix B in [46] and Appendix G in [12, 13]. These results are shown in Section 5.1.

## 2.2 A unifying method for the large deviations at short time

Whenever a Pfaffian or determinantal representation exists for the moment generating function of the partition function $Z = e^{H_1}$ as in (7), we present a general method and general mathematical rules to obtain the exact distribution of $H_1$ at short time. This method extends the ones used in [28] and [30] for the droplet and stationary initial conditions respectively in the full space.

*Result* 1 (Short time large deviations properties). We suppose that the KPZ equation has been solved and yields for the moment generating function of the partition function the following Fredholm Pfaffian point process representation for $z \geq 0$

$$\mathbb{E}_{\text{KPZ}}\left[\exp\left(-\frac{z\alpha}{\sqrt{t}}e^{H_1}\right)\right] = \mathbb{E}_K\left[\prod_{i=1}^{\infty}\frac{1}{[1+ze^{t^{1/3}a_i}]^\chi}\right], \tag{8}$$

for some $\alpha > 0$, $\chi > 0$ and a set of points $\{a_i\}_{i\in\mathbb{N}}$ forming a Pfaffian point process with a $2 \times 2$ kernel $(K_{ij})_{i,j=1,2}$. We suppose the following properties on the off-diagonal kernel

1. $K_{12}(at^{-1/3}, at^{-1/3}) \simeq_{t\ll 1} t^{-1/6}\rho_\infty(a)\theta(a \leq \Xi)$ for some finite $\Xi < \infty$ where $\theta$ is the Heaviside function.

2. $\rho_\infty$ is positive real-valued and strictly decreasing on $]-\infty, \Xi]$ and grows towards $-\infty$ as $\rho_\infty(a) \simeq_{-a\gg 1} \beta_1[-a]^{\gamma_1}$ for some $\beta_1 > 0$ and $\gamma_1 > 0$.

3. $\rho_\infty$ vanishes algebraically at the right edge $\Xi$ as $\rho_\infty(a) \simeq_{a\to\Xi} (\Xi - a)^\nu$ for some $0 < \nu \leq 1$.

4. The extension of $\rho_\infty$ on the interval $]\Xi, +\infty[$ is purely imaginary-valued and grows toward $+\infty$ as $\rho_\infty(a) \simeq_{a\gg 1} \beta_2[-a]^{\gamma_2}$ for some $\beta_2 > 0$ and $\gamma_2 > 0$. It requires $\gamma_2$ to be half-integer as discussed below.

We introduce the functions $\Psi$ defined on $[-e^{-\Xi}, +\infty[$ and $f$ defined on $]0, +\infty[$

$$\Psi(z) = \chi\int_{-\infty}^{\Xi} da\, \log(1 + ze^a)\rho_\infty(a), \qquad f(y) = \chi\int_{-\log y}^{\Xi} dv\, \rho_\infty(v). \tag{9}$$

Then, the random variable defined as $H = H_1 + \log\alpha - \log\Psi'(0)$ is centered, i.e. $\mathbb{E}[H] = 0$, and its probability density function takes the large deviation form at short time

$$\log P(H, t) \underset{t\ll 1}{\simeq} -\frac{\Phi(H)}{\sqrt{t}}. \tag{10}$$

Introducing the "branching field" $H_c = \log[\Psi'(-e^{-\Xi})/\Psi'(0)]$, see Section 4.2, $\Phi$ is the solution of the implicit equations

$$
\begin{aligned}
\forall H \leq H_c, \ \Phi(H) - \Phi'(H) &= \Psi\left(-\frac{e^{-H}\Phi'(H)}{\Psi'(0)}\right), \\
\forall H \geq H_c, \ \Phi(H) - \Phi'(H) &= \Psi\left(-\frac{e^{-H}\Phi'(H)}{\Psi'(0)}\right) + 2i\pi f\left(\frac{e^{-H}\Phi'(H)}{\Psi'(0)}\right),
\end{aligned}
\tag{11}
$$

or equivalently of the parametric equations

- $\forall H \leq H_c$, or equivalently, $\forall z \in [-e^{-\Xi}, +\infty[$ (see Section 4.1)

$$
\begin{aligned}
\Psi'(0)e^H &= \Psi'(z), \\
\Phi(H) &= \Psi(z) - z\Psi'(z).
\end{aligned}
\tag{12}
$$

- $\forall H \geq H_c$, or equivalently, $\forall z \in [-e^{-\Xi}, 0[$ (see Section 4.2)

$$
\begin{aligned}
\Psi'(0)e^H &= \Psi'(z) - 2i\pi f'(-z), \\
\Phi(H) &= \Psi(z) - z\Psi'(z) + 2i\pi f(-z) + 2i\pi z f'(-z).
\end{aligned}
\tag{13}
$$

The properties of $\Phi$ are the following :

1. $\Phi$ is analytic, i.e. infinitely differentiable everywhere on the real line, see Section 4.8.

2. $\Phi$ is quadratic for small argument, i.e. $\Phi(0) = \Phi'(0) = 0$, $\Phi''(0) = -\Psi'(0)^2/\Psi''(0)$ so that the distribution of $H$ is gaussian for small $H$ around 0. The second cumulant of $H$ is $\mathbb{E}[H^2] = -\frac{\Psi''(0)}{\Psi'(0)^2}\sqrt{t}$ and the higher ones are provided in Section 4.3 Eq. (45).

3. The left tail of $\Phi$ is $\Phi(H) \simeq_{H \to -\infty} \dfrac{\chi \beta_1}{(\gamma_1 + 1)(\gamma_1 + 2)}|H|^{\gamma_1 + 2}$, see Section 4.6.

4. The right tail of $\Phi$ is $\Phi(H) \simeq_{H \to +\infty} \dfrac{2\pi \chi \beta_2}{\gamma_2 + 1}H^{\gamma_2 + 1}$, see Section 4.7.

*Remark* 2.1 (Determinantal point process). This unifying method is also valid for a determinantal point process with Kernel $K$ up to the replacement of $K_{12}(at^{-1/3}, at^{-1/3})$ in the Hypothesis 1 by $K(at^{-1/3}, at^{-1/3})$.

*Remark* 2.2 (Dynamical phase transition). Under our set of hypothesis, as $\Phi$ is analytic, there cannot be a dynamical phase transition for the large deviation statistics. In Ref. [30], it was shown exactly for the Brownian initial condition that $\Phi$ exhibits a singularity in its second derivative. The reason for that is that in the Brownian case, the Hypothesis 4 about the growth of $\rho_\infty$ is violated. This phase transition was unveiled in the context of WNT [24, 26].

This very general framework is then applied to specific examples: we summarize below the results and properties obtained for the half-space droplet KPZ solution for $A = +\infty, -\frac{1}{2}, 0$ respectively in Sections 5, 6 and 7. The general features have been discussed in the Introduction and the details are summarized in the following Table 2.

Table 2: Summary of results

| Properties | $A = +\infty$ | $A = 0$ | $A = -\frac{1}{2}$ |
|---|---|---|---|
| Fermi factor power $\chi$ | 1 | 1 | $\frac{1}{2}$ |
| $\rho_\infty(a)$ | $\frac{1}{2\pi}\left[\sqrt{-W_{-1}(-e^a)} - \sqrt{-W_0(-e^a)}\right]$ | $\frac{1}{2\pi}\sqrt{-a}$ | $\frac{1}{\pi}\sqrt{-a}$ |
| Edge $\Xi$ | -1 | 0 | 0 |
| Left asymptotics of $\rho_\infty$ | $\frac{1}{2\pi}\sqrt{-a}$ | $\frac{1}{2\pi}\sqrt{-a}$ | $\frac{1}{\pi}\sqrt{-a}$ |
| Edge cancellation of $\rho_\infty$ | $\sqrt{-1-a}$ | $\sqrt{-a}$ | $\sqrt{-a}$ |
| Right asymptotics of $\rho_\infty$ | $i\frac{1}{\pi}\sqrt{a}$ | $i\frac{1}{2\pi}\sqrt{a}$ | $i\frac{1}{\pi}\sqrt{a}$ |
| Centered field $H$ | $h(\varepsilon,t) + \frac{t}{12} - \log\frac{\varepsilon^2}{\sqrt{4\pi t^{3/2}}} \quad (\varepsilon \to 0^+)$ | $h(0,t) + \frac{t}{12} + \frac{1}{2}\log(\pi t)$ | |
| Second cumulant $\mathbb{E}[H^2]^c$ | $\frac{3}{2}\sqrt{\frac{\pi t}{2}}$ | $\sqrt{2\pi t}$ | |
| Third cumulant $\mathbb{E}[H^3]^c$ | $\left(\frac{160}{27\sqrt{3}} - \frac{27}{8}\right)\pi t$ | $\frac{2}{9}(16\sqrt{3} - 27)\pi t$ | |
| Fourth cumulant $\mathbb{E}[H^4]^c$ | $\frac{5}{144}\left(567 + 486\sqrt{2} - 512\sqrt{6}\right)\pi^{3/2}t^{3/2}$ | $\frac{8}{3}\left(18 + 15\sqrt{2} - 16\sqrt{6}\right)\pi^{3/2}t^{3/2}$ | |
| Fifth cumulant $\mathbb{E}[H^5]^c$ | $\left(-\frac{10296145}{23328} - \frac{4725}{8\sqrt{2}} + 400\sqrt{3} + \frac{1161216}{3125\sqrt{5}}\right)\pi^2 t^2$ | $\frac{8}{225}(-39625 - 27000\sqrt{2} + 36000\sqrt{3} + 6912\sqrt{5})\pi^2 t^2$ | |
| Branching field $H_c$ | 0.9795 | 0.9603 | |
| Left tail of $\Phi$ | $\frac{2}{15\pi}|H|^{5/2}$ | $\frac{2}{15\pi}|H|^{5/2}$ | |
| Right tail of $\Phi$ | $\frac{4}{3}H^{3/2}$ | $\frac{2}{3}H^{3/2}$ | |
| Is $\Phi$ analytic ? | yes | yes | |

Note that $W_0$ and $W_{-1}$ are the two real branches of the Lambert function, i.e. $W(z)e^{W(z)} = z$, see Appendix A for more details and [47] for a review about the Lambert function.

# 3 Large deviation of the moment generating function from the first cumulant

## 3.1 Introduction to the cumulant method

Throughout this section we assume that the KPZ equation has been solved and yields for the moment generating function of the partition function the following Fredholm Pfaffian representation for $z \geq 0$

$$\mathbb{E}_{\text{KPZ}}\left[\exp\left(-\frac{z\alpha}{\sqrt{t}}e^H\right)\right] = \mathbb{E}_K\left[\prod_{i=1}^{\infty}\frac{1}{[1 + ze^{t^{1/3}a_i}]^{\chi}}\right], \tag{14}$$

for some $\alpha > 0$, $\chi > 0$ and properly shifted height field $H$, where the set $\{a_i\}_{i \in \mathbb{N}}$ forms a Pfaffian point process

$$\mathbb{E}_K \left[ \prod_{i=1}^{\infty} \frac{1}{[1 + z e^{t^{1/3} a_i}]^\chi} \right] = \mathrm{Pf}[J - \sigma_{t,z} K], \qquad (15)$$

with the $2 \times 2$ kernels $K$ and $J$, the generalized Fermi factor $\sigma_{t,z}$ being defined as

$$J(r, r') = \begin{pmatrix} 0 & 1 \\ -1 & 0 \end{pmatrix} \mathbb{1}_{r=r'}, \qquad K = \begin{pmatrix} K_{11} & K_{12} \\ K_{21} & K_{22} \end{pmatrix}, \qquad \sigma_{t,z}(a) = 1 - \frac{1}{[1 + z e^{t^{1/3} a}]^\chi}. \quad (16)$$

$K$ should be anti-symmetric, therefore $K_{21}(r, r') = -K_{12}(r', r)$. The expectation value on the l.h.s of (14) is taken over the realization of the KPZ white noise and the one on the r.h.s of (14) is taken over the Pfaffian point process.

*Remark* 3.1. The coefficient $\alpha$ accounts for an eventual shift in the solution of the KPZ equation to center its distribution around 0.

*Remark* 3.2. For the solved cases of the KPZ equation which fulfill the Pfaffian representation (14), we have $\chi = 1$ or $\chi = \frac{1}{2}$.

Introducing the function $\varphi_{t,z}(a) = \chi \log(1 + z e^{t^{1/3} a})$ (and subsequently dropping the subscript), using the identities $\mathrm{Pf}[J - \sigma K]^2 = \mathrm{Det}[1 + \sigma J K]$, $\log \mathrm{Det} = \mathrm{Tr} \log$ and series expanding $\log(1-x)$, we write the logarithm of (15) as

$$\log \mathbb{E}_K \left[ \exp\left( -\sum_{i=1}^{\infty} \varphi(a_i) \right) \right] = -\frac{1}{2} \sum_{p=1}^{\infty} \frac{1}{p} \mathrm{Tr}\left[ (e^{-\varphi} - 1) J K \right]^p. \qquad (17)$$

Expanding this series in powers of $\varphi$ leads to the cumulant expansion of the Pfaffian

$$-\frac{1}{2} \sum_{p=1}^{\infty} \frac{1}{p} \mathrm{Tr}\left[ (e^{-\varphi} - 1) J K \right]^p = \sum_{n=1}^{\infty} \frac{\kappa_n}{n!}. \qquad (18)$$

where the $n$-th cumulant $\kappa_n$ is defined as $n!$ times the term of order $\varphi^n$ in this expansion. The idea to introduce the cumulant expansion at short time originates from Refs. [36] where it was observed that the first cumulant yields the entire large deviation function for the moment generating function (14), previously calculated in [28, 30] for the droplet and stationary ICs in full-space, through an involved resummation of traces arising from expanding Fredholm determinants. Here and below, we follow this approach that we call the *cumulant approximation* which states that at short time

$$\log \mathbb{E}_K \left[ \exp\left( -\sum_{i=1}^{\infty} \varphi(a_i) \right) \right] \underset{t \ll 1}{\simeq} \kappa_1. \qquad (19)$$

The validity of this approximation can be understood as follows. As seen from Eq. (17), the cumulants $\kappa_n$ are the ones of the random variable $X = \sum_{i=1}^{\infty} \varphi(a_i)$ where $\varphi(a) = \chi \log(1 + z e^{t^{1/3} a})$ and the set $\{a_i\}$ forms a Pfaffian point process. In the limit $t \ll 1$ many of the $a_i$'s contribute to the sum, and by a law of large number, the fluctuations of $X$ around the mean value are subdominant. This is confirmed by an explicit calculation of higher order cumulants $n \geq 2$ in Ref. [48] where cancellations occur leaving only subdominant

powers of $t$.

The first cumulant reads

$$
\begin{aligned}
\kappa_1 &= \frac{1}{2}\mathrm{Tr}(\varphi J K) = -\mathrm{Tr}(\varphi K_{12}) \\
&= -\chi \int_{\mathbb{R}} \mathrm{d}a\, \log(1 + z e^{t^{1/3}a})K_{12}(a,a).
\end{aligned}
\tag{20}
$$

We define the density $\rho(a) = K_{12}(a,a)$ and rescale the integration variable by $t^{-1/3}$

$$
\kappa_1 = -\frac{\chi}{t^{1/3}} \int_{\mathbb{R}} \mathrm{d}a\, \log(1 + z e^a)\rho(a t^{-1/3}).
\tag{21}
$$

As stated in Section 2.2, we suppose the following properties on the asymptotic density

- $\rho(a t^{-1/3}) \simeq_{t \ll 1} t^{-1/6}\rho_\infty(a)\theta(a \le \Xi)$ for some finite $\Xi < \infty$ where $\theta$ is the Heaviside function.

- $\rho_\infty$ is positive real-valued and strictly decreasing on $]-\infty, \Xi]$ and grows towards $-\infty$ as $\rho_\infty(a) \simeq_{-a \gg 1} \beta_1[-a]^{\gamma_1}$ for some $\beta_1 > 0$ and $\gamma_1 > 0$.

- $\rho_\infty$ vanishes algebraically at the right edge $\Xi$ as $\rho_\infty(a) \simeq_{a \to \Xi} (\Xi - a)^\nu$ for some $\nu > 0$.

- The extension of $\rho_\infty$ on the interval $]\Xi, +\infty[$ is purely imaginary-valued and grows toward $+\infty$ as $\rho_\infty(a) \simeq_{a \gg 1} \beta_2[-a]^{\gamma_2}$ for some $\beta_2 > 0$ and $\gamma_2 > 0$. It requires $\gamma_2$ to be half-integer as discussed below.

*Remark* 3.3. The reason for the extension of $\rho_\infty$ to be purely imaginary valued above $\Xi$ comes from its derivation from the off-diagonal kernel element $K_{12}$. In the cases studied, $K_{12}$ is be defined through a contour integral in the complex plane and $\rho_\infty(a)$ will be given by the saddle point of the integrand at short time. The threshold $\Xi$ will be defined by the frontier where $\rho_\infty(a)$ turns from being real strictly positive to being purely imaginary. In our cases of interest, the fact that $\rho_\infty(a)$ becomes purely imaginary corresponds to an exponential decay for the kernel $K_{12}(a t^{-1/3}, a t^{-1/3})$ for $a \ge \Xi$, which we can approximate by a $\theta$ function for the density at short time.

## 3.2 Large deviation of the moment generating function

Making use of the properties of $\rho_\infty$ stated in Section 3.1, we are now able to introduce the large deviation expression $\kappa_1 = -\frac{\Psi(z)}{\sqrt{t}}$ with $\Psi$ defined as

$$
\Psi(z) = \chi \int_{-\infty}^{\Xi} \mathrm{d}a\, \log(1 + z e^a)\rho_\infty(a).
\tag{22}
$$

Defining the integrated density $f(y) = \chi \int_{-\log y}^{\Xi} \mathrm{d}v\, \rho_\infty(v)$ and the strictly positive variable $\zeta = e^{-\Xi}$, we rewrite $\Psi$ using an integration by part and a change of variable $y = e^{-a}$

$$
\Psi(z) = \int_{\zeta}^{+\infty} \mathrm{d}y\, f(y)\frac{z}{y}\frac{1}{y+z}.
\tag{23}
$$

To summarize, the *cumulant approximation* allows to introduce the Large Deviation Principle for the moment generating function (14) for $z \ge 0$

$$
\log \mathbb{E}_{\mathrm{KPZ}}\left[\exp\left(-\frac{z\alpha}{\sqrt{t}}e^H\right)\right] \underset{t \ll 1}{\simeq} -\frac{\Psi(z)}{\sqrt{t}}.
\tag{24}
$$

*Remark* 3.4. The moment generating function, i.e. the l.h.s. of (24) is infinite for $z < 0$, hence (24) holds only for $z \geq 0$. The function $\Psi(z)$ however is also defined for some negative values of $z$ (i.e. in the interval $z \in I = [-\zeta, +\infty[$ see below). Accordingly (24) also holds as a power series in $z$ around $z = 0$ (and allows to extract the moments $\mathbb{E}_{\mathrm{KPZ}}[e^{nH}]$ see below).

### 3.3 Analytic properties of $f$ and $\Psi$

From the definitions of $f$ and $\Psi$ in Eq. (22), one deduces some analytic properties

- $f(\zeta) = f'(\zeta) = \Psi(0) = 0$.

- $f$ is purely imaginary-valued on the interval $]0, \zeta]$.

- $\Psi$ is defined on the interval $I = [-\zeta, +\infty[$, is strictly increasing and strictly concave on $I$ and is infinitely differentiable on $]-\zeta, +\infty[$.

- $\Psi$ has a branch cut in the complex plane along the $]-\infty, -\zeta[$ axis.

- Recalling that $\rho_\infty$ vanishes algebraically as $\rho_\infty(a) \simeq_{a \to \Xi} (\Xi - a)^\nu$ and defining $n$ the least integer greater than $\nu$, then for all $k \leq n$, $\Psi^{(k)}(-\zeta)$ are finite, and for all $k > n$, $\Psi^{(k)}(-\zeta)$ are infinite. The reason for this is that

$$\Psi^{(k)}(-\zeta) = \chi(-1)^{k-1}(k-1)! \int_{-\infty}^{\Xi} da \, \frac{e^{ka}}{(1 - e^{a-\Xi})^k} \rho_\infty(a). \tag{25}$$

Expanding the integrand near the right edge as $a = \Xi - \varepsilon$, we obtain

$$\frac{e^{ka}}{(1 - e^{a-\Xi})^k} \rho_\infty(a) \underset{a = \Xi - \varepsilon}{\simeq} e^{k(\Xi - \varepsilon)} \varepsilon^{\nu - k}, \tag{26}$$

which is integrable if $\nu + 1 > k$. In particular, as $\nu$ is strictly positive, $\Psi(-\zeta)$ and $\Psi'(-\zeta)$ are finite.

*Remark* 3.5. In all studied cases, we have $\nu \leq 1$, hence only the first two orders of $\Psi$ are finite, i.e. $\Psi(-\zeta) < \infty, \Psi'(-\zeta) < \infty$ and $\Psi''(-\zeta) = \infty$.

### 3.4 Asymptotics of $\Psi$ at $z \to +\infty$

We investigate the asymptotic properties of $\Psi$ for large positive argument starting from (23). As it will be discussed in Section 4.6, these asymptotics provide the left tail of the distribution of the KPZ solution.

$$\Psi(z) = \int_\zeta^{+\infty} dy \, \frac{f(y)}{y} \frac{1}{\frac{y}{z} + 1}. \tag{27}$$

The denominator of the integrand is close to one for $y \ll z$ and very large for $y \gg z$ which suggests splitting the range of integration at $y = z$, giving

$$\Psi(z) = \int_\zeta^z dy \, \frac{f(y)}{y} - \int_\zeta^z dy \, \frac{f(y)}{y} \frac{1}{\frac{z}{y} + 1} + \int_z^{+\infty} dy \, \frac{f(y)}{y} \frac{1}{\frac{y}{z} + 1}. \tag{28}$$

Similarly to the computation of the asymptotics of the polylogarithm function [49] one shows that the first integral is the leading term for large argument

$$\Psi(z) \underset{z \to +\infty}{\simeq} \int_\zeta^z dy \, \frac{f(y)}{y} = \chi \int_{-\Xi}^{\log z} dr \int_{-\Xi}^r dv \, \rho_\infty(-v). \tag{29}$$

Recalling that $\rho_\infty$ has a polynomial growth for large negative argument $\rho_\infty(v) \simeq_{-v \gg 1} \beta_1[-v]^{\gamma_1}$ for some $\beta_1 > 0$ and $\gamma_1 > 0$, the integral (29) is asymptotically equal to

$$\Psi(z) \underset{z \to +\infty}{\simeq} \frac{\chi \beta_1}{(\gamma_1 + 1)(\gamma_1 + 2)}[\log z]^{\gamma_1 + 2}. \tag{30}$$

*Remark* 3.6. For all observed cases, we found that $\rho_\infty$ has a square root divergence for large negative argument, $\rho_\infty(a) \simeq_{-a \gg 1} \beta_1 \sqrt{|a|}$, i.e $\gamma_1 = \frac{1}{2}$, hence $\Psi(z) \underset{z \to +\infty}{\simeq} \frac{4\chi\beta_1}{15}[\log z]^{\frac{5}{2}}$.

## 3.5 Analytic continuation of $\Psi$

As $\Psi$ exhibits a branch cut along the interval $]-\infty, -\zeta[$, one can define its extension from the complex plane to a Riemann surface. Starting from the formulation (23) $\Psi(z) = \int_\zeta^{+\infty} dy f(y) \frac{z}{y} \frac{1}{y+z}$, one uses the following expression that makes sense in distribution theory to study the jump of $\Psi$ across the branch cut $]-\infty, -\zeta[$

$$\lim_{\epsilon \to 0} \frac{1}{y + z \pm i\epsilon} = \mathscr{P}(\frac{1}{y+z}) \mp i\pi\delta(-z), \tag{31}$$

and defines $\Delta$ to be the jump of $\Psi$ across the branch cut $]-\infty, -\zeta[$.

$$\Delta(z) = \lim_{\epsilon \to 0}[\Psi(z + i\epsilon) - \Psi(z - i\epsilon)] = 2i\pi f(-z). \tag{32}$$

We consequently define the continuation of $\Psi$ as a multi-valued function $\Psi_{\text{continued}}$ defined on $[-\zeta, 0[$ and consider the multi-valuation as the projection of $\Psi$, viewed as a function on a Riemann surface, onto the complex plane.

$$\forall z \in [-\zeta, 0[, \quad \Psi_{\text{continued}}(z) = \Psi(z) + 2i\pi f(-z). \tag{33}$$

The imaginary valuation of $f$ on $]0, \zeta]$ makes sense as $\Psi_{\text{continued}}$ should be real-valued for physical reasons. The regularity of the continuation $\Psi \to \Psi_{\text{continued}}$ will be controlled by the behavior of $f$ around $\zeta$.

## 3.6 Behavior of $f$ for small positive argument

We investigate the function $f$ for an argument in the interval $]0, \zeta[$ and more particularly, we will characterize the possible divergence of $f$ for small positive argument which provides the right tail of the distribution of the KPZ equation as discussed in Section 4.7.

$$f(y) = \chi \int_{-\log y}^{\Xi} dv \, \rho_\infty(v). \tag{34}$$

Recalling that the extension of $\rho_\infty$ on the interval $]\Xi, +\infty[$ is purely imaginary-valued and grows toward $+\infty$ as $\rho_\infty(a) \simeq_{a \gg 1} \beta_2[-a]^{\gamma_2}$ for some $\beta_2 > 0$ and $\gamma_2 > 0$, then for small positive argument $y$, $f$ will asymptotically be equal to

$$f(y) \underset{y \to 0^+}{\simeq} \frac{\chi \beta_2}{\gamma_2 + 1}[\log y]^{\gamma_2 + 1}. \tag{35}$$

To be purely imaginary, we also require $\gamma_2$ to be a half-integer of the form $m + \frac{1}{2}$ so that

$$f(y) \underset{y \to 0^+}{\simeq} i(-1)^{1+m} \frac{\chi \beta_2}{\frac{3}{2} + m}[-\log y]^{\frac{3}{2} + m}. \tag{36}$$

*Remark* 3.7. As $\gamma_2 > 0$, the logarithmic divergence of $f$ will be at least of magnitude $\dfrac{3}{2}$.

*Remark* 3.8. If $m$ is odd, we will define the jump of $\Psi$ to be $-\Delta$ instead of $\Delta$, accounting for a jump to the lower Riemann sheet instead of the upper Riemann sheet. Therefore, the factor $(-1)^m$ can be fully ignored.

If $f$ does not exhibit a divergence for small argument, because of the branch cut in the lower boundary of the integral (34) along the real negative axis, further investigation will be required on the case by case basis to obtain additional properties on $f$.

# 4 Inverting the moment generating function: a general method

## 4.1 General framework

Let $H$ be the solution of the KPZ equation such that for short times $t \ll 1$ we determined the Large Deviation Principle (24), i.e. $\log \mathbb{E}_{\text{KPZ}}\left[\exp\left(-\frac{z\alpha}{\sqrt{t}}e^H\right)\right] \simeq_{t \ll 1} -\frac{\Psi(z)}{\sqrt{t}}$ for $z \geq 0$. We impose the density $P(H, t)$ at short time to be of the form $\log P(H, t) \simeq_{t \ll 1} -\frac{\Phi(H)}{\sqrt{t}}$ yielding the large deviation estimate of the moment generating function for $z \geq 0$

$$\log \mathbb{E}_{\text{KPZ}}\left[\exp\left(-\frac{z\alpha}{\sqrt{t}}e^H\right)\right] \underset{t \ll 1}{\simeq} \log \int_{\mathbb{R}} dH \exp\left[-\frac{1}{\sqrt{t}}\left(z\alpha e^H + \Phi(H)\right)\right]. \tag{37}$$

Using $1/\sqrt{t}$ as a large parameter, the integral in the r.h.s of (37) can be evaluated by a saddle point method. It gives for $z \geq 0$

$$\Psi(z) = \min_{H \in \mathbb{R}}\left[z\alpha e^H + \Phi(H)\right]. \tag{38}$$

and one can invert the resulting Legendre transform to obtain the large deviation rate function $\Phi$ as the solution of an optimization problem

$$\Phi(H) = \max_{z \in I}\left[\Psi(z) - z\alpha e^H\right]. \tag{39}$$

*Remark* 4.1. As $\Psi$ is strictly concave, (39) has a unique solution.

*Remark* 4.2. As $\Phi$ is the large deviation rate function for a real random variable $H$ centered around 0, we impose the two properties $\Phi(0) = 0$ and $\Phi'(0) = 0$.

We now solve the optimization problem (39) either parametrically or implicitly

- The parametric solution is obtained by differentiating (39) w.r.t to $z$ and re-injecting the optimal $z$ in the optimization equation.

$$\begin{cases} \alpha e^H = \Psi'(z) \\ \Phi(H) = \Psi(z) - z\Psi'(z). \end{cases} \tag{40}$$

- One can further invert the relation between $H$ and $z$ by taking the total derivative w.r.t $H$ of (39). One then obtains $z = -\frac{\Phi'(H)e^{-H}}{\alpha}$ and the implicit solution

$$\Phi(H) - \Phi'(H) = \Psi\left(-\frac{e^{-H}\Phi'(H)}{\alpha}\right), \tag{41}$$

which is the result announced in Section 2.2 Eqs. (11) and (12).

*Remark* 4.3. The parametric solution is quite useful to plot $\Phi$ while the implicit solution is useful to derive the small argument expansion and large argument asymptotics of $\Phi$.

*Remark* 4.4. While the moment generating function (24) is defined for $z \geq 0$ a priori, we extend the solution of the optimization problem (39) to $z \in I = [-\zeta, +\infty[$.

## 4.2 Range of solution of the optimization problem and continuation of $\Phi$

Starting from the parametric representation of the field $H$, $\alpha e^H = \Psi'(z)$, using the decrease of $\Psi'$ on $I$ from $\Psi'(-\zeta)$ to $\Psi'(+\infty) = 0$, one sees that the parametric solution (40) allows to obtain $\Phi(H)$ for $H \in \left]-\infty, \log\left[\frac{\Psi'(-\zeta)}{\alpha}\right]\right]$.

Furthermore, imposing the distribution of $H$ to be centered around 0, i.e. $\Phi'(0) = 0$, and using the relation $z = -\frac{\Phi'(H)e^{-H}}{\alpha}$, one sees that $H = 0$ corresponds to $z = 0$. As a consequence, the value of $\alpha$ is determined by $\alpha = \Psi'(0)$.

Under this centering constraint, the critical value of $H$ below which a solution of the optimization problem (39) exists is $H_c = \log\frac{\Psi'(-\zeta)}{\Psi'(0)}$. It is strictly positive as $\zeta$ is strictly positive and $\Psi'$ is strictly decreasing.

*Remark* 4.5. There is an ambiguity in the relation $z = -\frac{\Phi'(H)e^{-H}}{\Psi'(0)}$, where $z = 0$ could correspond either to $H = 0$ or $H = +\infty$. This ambiguity is lifted when considering the multi-valuation of $\Psi$ or equivalently, the multi-valuation of $H(z) = \log\frac{\Psi'(z)}{\Psi'(0)}$. Note that this does not contradict the uniqueness of $\Phi(H)$.

We may wonder what are the consequences of solving the optimization problem (39) only in the range $H \in \left]-\infty, H_c\right]$, and how one can obtain the remaining part of the distribution for $H \in [H_c, +\infty[$. To answer these questions, we extend the optimization problem (39), or equivalently, its solutions (40) and (41) by proceeding to the minimal replacement $\Psi \to \Psi_{\text{continued}}$, where $\Psi_{\text{continued}}$ is defined in Section 3.5 Eq. (33) as

$$\forall z \in [-\zeta, 0[, \quad \Psi_{\text{continued}}(z) = \Psi(z) + 2i\pi f(-z). \tag{42}$$

*Remark* 4.6. The interpretation of this replacement is that either we consider from the beginning $\Psi$ to be defined on a Riemann surface and then the optimization problem (39) has to be considered over this surface, or we separate the resolution of this optimization problem for each valuation $\Psi$ over the real line.

*Remark* 4.7. As $\Psi$ is the large deviation representation of the moment generating function (14), considering $\Psi$ on a Riemann surface is equivalent to considering the moment generating function on the same surface. This feature is quite unusual and it is the first time to our knowledge it does appear in the literature. We conjecture this to be related to the *moment problem* in probability, see [50].

The continued version of the parametric solution (40) reads

$$\begin{cases} \Psi'(0)e^H = \Psi'(z) - 2i\pi f'(-z) \\ \Phi(H) = \Psi(z) - z\Psi'(z) + 2i\pi f(-z) + 2i\pi z f'(-z). \end{cases} \tag{43}$$

The continued version of the implicit solution (41) reads

$$\Phi(H) - \Phi'(H) = \Psi\left(-\frac{e^{-H}\Phi'(H)}{\Psi'(0)}\right) + 2i\pi f\left(\frac{e^{-H}\Phi'(H)}{\Psi'(0)}\right), \tag{44}$$

which is the result announced in Section 2.2 Eqs. (11) and (13). The regularity of this continuation will be discussed in Section 4.8. The proof that (43) and (44) allow to obtain $\Phi(H)$ for $H > H_c$ is left on the case by case basis where the relation $\Psi'(0)e^H = \Psi'(z) - 2i\pi f'(-z)$ has to be interpreted. A schematic representation of the parametric solution is presented in Fig. 1.

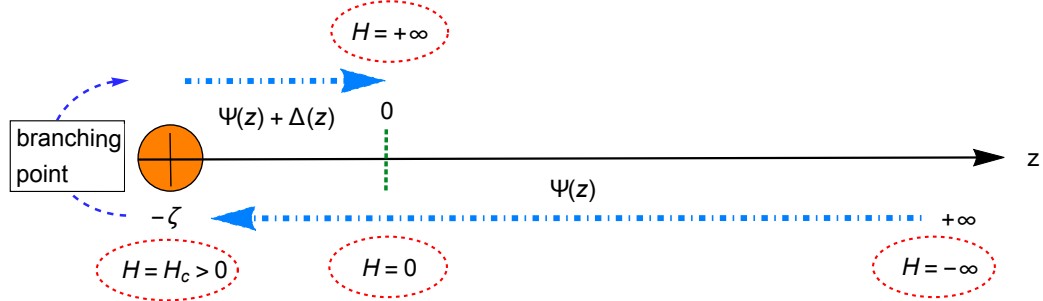

Figure 1: Schematic representation of the parametric solution of the optimization problem. For $H \leq H_c$ one uses the function $\Psi$ in the parametric representation (40) taking the parameter $z$ to decrease from $+\infty$ to $-\zeta$. At $H = H_c$ or $z = -\zeta$, one needs to turn around the branching point and replace $\Psi$ by its continuation $\Psi + \Delta$ in (43) to determine all $H \geq H_c$ by increasing the parameter $z$ from $-\zeta$ to 0.

### 4.3 Expansion for small $H$ and centering

The branching field $H_c$ being strictly positive, the derivatives of $\Phi$ at $H = 0$ are well defined and one can expand (41) in $H$ to obtain the derivatives of $\Phi$ which we provide up to the fifth order

$$\Phi^{(2)}(0) = -\frac{\Psi'(0)^2}{\Psi''(0)}, \qquad \Phi^{(3)}(0) = \frac{\Psi^{(3)}(0)\Psi'(0)^3 - 3\Psi'(0)^2\Psi''(0)^2}{\Psi''(0)^3},$$

$$\Phi^{(4)}(0) = \frac{3\Psi^{(3)}(0)^2 - \Psi^{(4)}(0)\Psi''(0)}{\Psi''(0)^5}, \tag{45}$$

$$\Phi^{(5)}(0) = \frac{-15\Psi^{(3)}(0)^3 - \Psi^{(5)}(0)\Psi''(0)^2 + 10\Psi^{(4)}(0)\Psi^{(3)}(0)\Psi''(0)}{\Psi''(0)^7}.$$

In order to center the variable $H$, assume that initially, we obtained the moment generating function as $\log \mathbb{E}_{\text{KPZ}}\left[\exp\left(-z\tilde{\alpha}e^{H_1}\right)\right]$, defining $H = H_1 + \log\tilde{\alpha} - \log\Psi'(0) + \frac{1}{2}\log t$, the moment generating function expressed in terms of $H$ is $\mathbb{E}_{\text{KPZ}}\left[\exp\left(-\frac{z\Psi'(0)}{\sqrt{t}}e^H\right)\right]$ and therefore, according to Section 4.2 and as stated in Section 2.2, $H$ is centered around 0.

### 4.4 Cumulants of the partition function

Defining the partition function $Z = e^H$, we express the moment generating function of $Z$ in terms of its cumulant expansion

$$\log \mathbb{E}\left[\exp\left(-\frac{z\Psi'(0)Z}{\sqrt{t}}\right)\right] = \sum_{q=1}^{\infty} \frac{\mathbb{E}\left[Z^q\right]^c}{q!}\left(-\frac{z\Psi'(0)}{\sqrt{t}}\right)^q. \tag{46}$$

Using the large deviation expression (24) and expanding $\Psi$ in terms of its Taylor series around 0, $\Psi(z) = \sum_{q=1}^{\infty}\frac{z^q}{q!}\Psi^{(q)}(0)$, we express the $q$-th cumulant of $Z$ as

$$\mathbb{E}\left[Z^q\right]^c = (-1)^{q+1}\frac{\Psi^{(q)}(0)}{\Psi'(0)^q}t^{\frac{q-1}{2}}. \tag{47}$$

### 4.5 Cumulants of the height field

Similarly to the computation of the cumulants of the partition function, one can compute the cumulants of the height field, see Refs. [28,30]. Indeed, the cumulant expansion $\phi$ is defined

as $\mathbb{E}_{\text{KPZ}}\left[e^{\frac{pH}{\sqrt{t}}}\right] = e^{\frac{\phi(p)}{\sqrt{t}}}$ and a saddle point expansion at short time yields

$$\phi(p) = \max_{H \in \mathbb{R}}\left[pH - \Phi(H)\right]. \tag{48}$$

By definition, $\mathbb{E}[H^q]^c = t^{\frac{q-1}{2}}\phi^{(q)}(0)$ and the $q$-th derivative of $\phi$ can be obtained by solving (48) as $\phi^{(q+1)}(0) = \left[\dfrac{1}{\Phi''(H)}\dfrac{\mathrm{d}}{\mathrm{d}H}\right]^q H\mid_{H=0}$ for all $q \geq 0$. The first five non trivial cumulants are given by

$$\phi^{(2)}(0) = \frac{1}{\Phi''(0)}, \qquad \phi^{(3)}(0) = -\frac{\Phi^{(3)}(0)}{\Phi''(0)^3}, \qquad \phi^{(4)}(0) = \frac{3\Phi^{(3)}(0)^2 - \Phi^{(4)}(0)\Phi''(0)}{\Phi''(0)^5},$$
$$\phi^{(5)}(0) = \frac{-15\Phi^{(3)}(0)^3 - \Phi^{(5)}(0)\Phi''(0)^2 + 10\Phi^{(4)}(0)\Phi^{(3)}(0)\Phi''(0)}{\Phi''(0)^7}. \tag{49}$$

*Remark* 4.8. Note that the equations (45) match the equations (90) of [30] in the limit $\tilde{w} \to \infty$ and the equations (49) match the equations (106) of [30].

In terms of the rate function $\Psi$, combining (45) and (49) we obtain

$$\phi^{(2)}(0) = -\frac{\Psi''(0)}{\Psi'(0)^2}, \qquad \phi^{(3)}(0) = \frac{\Psi^{(3)}(0)\Psi'(0) - 3\Psi''(0)^2}{\Psi'(0)^4},$$
$$\phi^{(4)}(0) = -\frac{20\Psi''(0)^3 + \Psi^{(4)}(0)\Psi'(0)^2 - 12\Psi^{(3)}(0)\Psi'(0)\Psi''(0)}{\Psi'(0)^6},$$
$$\phi^{(5)}(0) = \frac{-210\Psi''(0)^4 + \Psi^{(5)}(0)\Psi'(0)^3 + 180\Psi^{(3)}(0)\Psi'(0)\Psi''(0)^2}{\Psi'(0)^8}$$
$$\qquad\qquad - \frac{5\Psi'(0)^2\left(3\Psi^{(3)}(0)^2 + 4\Psi^{(4)}(0)\Psi''(0)\right)}{\Psi'(0)^8}. \tag{50}$$

*Remark* 4.9. As $\Psi''(0) < 0$, by concavity of $\Psi$, we verify that the second cumulant is indeed positive which is consistent with our mathematical construction.

## 4.6 Left tail of $\Phi$, $H \to -\infty$

Starting from the implicit representation (41), one uses the asymptotics of $\Psi$ determined in (30) to study the behavior of the factor $\Psi(-\frac{e^{-H}\Phi'(H)}{\Psi'(0)})$ for large negative $H$ knowing that $\Phi'(H) < 0$ for $H < 0$. As $\Psi$ exhibits logarithmic asymptotics for large positive argument, $\Phi$ exhibits a polynomial growth for large negative argument

$$\Phi(H) \underset{H \to -\infty}{\simeq} \frac{\chi\beta_1}{(\gamma_1 + 1)(\gamma_1 + 2)}|H|^{\gamma_1 + 2}, \tag{51}$$

where the different coefficients were introduced in Section 3.4. The asymptotics of $\Psi$ therefore provide the left tail of the KPZ solution as announced in Section 2.2.

## 4.7 Right tail of $\Phi$, $H \to +\infty$

Starting from the continued implicit representation (44), one uses the asymptotics of $f$ determined in (36) to study the behavior of the factor $2i\pi f(\frac{e^{-H}\Phi'(H)}{\Psi'(0)})$ for large positive $H$ knowing that $\Psi(0) = 0$. In the case where $f$ exhibits logarithmic asymptotics for small positive argument, as stated in Section 2.2, $\Phi$ exhibits a polynomial growth for large positive argument

$$\Phi(H) \underset{H \to +\infty}{\simeq} \frac{2\pi\chi\beta_2}{\frac{3}{2} + m}H^{\frac{3}{2} + m}, \tag{52}$$

where the different coefficients were introduced in Section 3.6. If $f$ does not exhibit a logarithmic divergence for small positive argument, additional effort will have to be done in the case by case basis to determine the right tail. An example of this situation is the stationary IC in full space [30] where two continuations of $f$ had to be defined, leading a more complex Riemann surface for $\Psi$ and a singular behavior of $\Phi$ on the branching point of these continuations.

## 4.8 Expansion of $\Phi$ around its continuation

Starting from the implicit (41) and the continued implicit (44) solutions, we determine which condition ensures the regularity of the continuation. We first expand the implicit solution (41) around $H = H_c$ and determine the left derivative expansion

$$
\Phi(H_c) = \Psi(-\zeta) + \zeta \Psi'(-\zeta),
$$

$$
\forall k \geq 1, \ \Phi^{(k)}(H_c) = \left[ \frac{\Psi'(z)}{\Psi''(z)} \frac{\mathrm{d}}{\mathrm{d}z} \right]^{k-1} \left( -z \Psi'(z) \right) \big|_{z=-\zeta}. \tag{53}
$$

*Remark* 4.10. Another way to see this relation between the derivatives is to differentiate the parametric relation $\alpha e^H = \Psi'(z)$ which yields $\mathrm{d}H = \frac{\Psi''(z)}{\Psi'(z)} \mathrm{d}z$.

*Remark* 4.11. Remarkably, $\Psi$ only needs to be $\mathscr{C}^1$ for $\Phi$ to have finite derivatives at all orders at $H = H_c$. Indeed by induction, if $\Psi''(-\zeta) = \infty$ then for all $k \geq 1$, $\Phi^{(k)}(H_c) = \zeta \Psi'(-\zeta) < \infty$.

*Remark* 4.12. Note that there is a second solution for the set of derivatives for arbitrary $z$ coming from the implicit equation (41). For $H = H_c$ it reads $\Phi^{(k)}(H_c) = \zeta \Psi'(-\zeta)$ which surprisingly is identical to (53).

One obtains the right derivative expansion by proceeding to the minimal replacement $\Psi \to \Psi_{\mathrm{continued}}$. We find a sufficient condition for the continuation to be infinitely smooth $\{\Psi''(-\zeta) = \infty, f(\zeta) = 0 \text{ and } f'(\zeta) = 0\}$. As discussed in Section 3.3, this sufficient condition is observed in all existing cases. If this condition is not met, the large deviation rate function $\Phi$ might encounter a singularity leading to a dynamical phase transition.

*Remark* 4.13. Additionally, as discussed in Section 3.3, $\Psi$ is infinitely differentiable on $]-\zeta, +\infty[$, therefore if the above sufficient condition is verified, $\Phi$ will be infinitely differentiable everywhere on the real line as announced in Section 2.2.

## 5 Hard wall $A = \infty$

Defining the field $H_1 = h(\varepsilon, t) + \frac{t}{12} - \log \varepsilon^2$ for $\varepsilon = 0^+$, we determine two new representations of the moment generating function $\mathbb{E}_{\mathrm{KPZ}}\left[ \exp\left( -z e^{H_1} \right) \right]$ starting from the results of Ref. [33].

### 5.1 New Fredholm Pfaffian expression for the solution to the hard wall

We start from Eqs. (19,21,23) of Ref. [33] with the definition of the moment generating function of the Cole-Hopf solution of the KPZ equation and the string-replicated moment $Z(n_s, z)$.

1. The moment generating function

$$
\mathbb{E}_{\mathrm{KPZ}}\left[ \exp\left( -z e^{H_1} \right) \right] = \sum_{n_s=0}^{\infty} \frac{1}{n_s!} Z(n_s, z). \tag{54}
$$

2. The string-replicated moments expressed with the reduced variables $X_{2p-1} = m_p + 2ik_p$ and $X_{2p} = m_p - 2ik_p$ for $p \in [1, n_s]$

$$Z(n_s, z) = \prod_{p=1}^{n_s} \sum_{m_p \geq 1} \int_{\mathbb{R}} \frac{dk_p}{2\pi} (-z)^{m_p} \frac{b_{m_p, k_p}}{4ik_p} e^{-tm_p k_p^2 + \frac{t}{12} m_p^3} \operatorname{Pf}\left[\frac{X_i - X_j}{X_i + X_j}\right]_{2n_s \times 2n_s}, \quad (55)$$

$$b_{k,m} = \prod_{q=0}^{m-1} (q^2 + 4k^2) = \frac{2k}{\pi} \sinh(2\pi k)\Gamma(m + 2ik)\Gamma(m - 2ik). \quad (56)$$

Using the variables $X_{2p}$ and $X_{2p-1}$ one further re-expresses $Z(n_s, z)$ as

$$Z(n_s, z) = \prod_{p=1}^{n_s} \sum_{m_p \geq 1} \int_{\mathbb{R}} \frac{dk_p}{2\pi} (-z)^{m_p} \frac{\sin(\frac{\pi}{2}(X_{2p} - X_{2p-1}))}{2\pi} \Gamma(X_{2p})\Gamma(X_{2p-1})$$

$$e^{\frac{t}{24}[X_{2p}^3 + X_{2p+1}^3]} \operatorname{Pf}\left[\frac{X_i - X_j}{X_i + X_j}\right]_{2n_s \times 2n_s}. \quad (57)$$

We introduce the Mellin-Barnes resummation expressed in its *Fermi* form along the contour $\tilde{C} = a + i\mathbb{R}$ for some $a \in ]0, 1[$ to substitute the summation over integers to an integral in the complex plane.

$$\sum_{m \geq 1} (-z)^m f(m) = -\int_{\mathbb{R}} dr \frac{z}{z + e^{-r}} \int_{\tilde{C}} \frac{dw}{2i\pi} e^{-wr} f(w). \quad (58)$$

Here and below we keep the definition of the reduced variables $X_{2p}$ and $X_{2p-1}$ up to the substitution $m \to w$ imposed by the Mellin-Barnes formula. We further proceed to the change of variable $(w_p, k_p) \to (X_{2p}, X_{2p-1})$ and define the contour $C = \frac{a}{2} + i\mathbb{R}$ so that the string-replicated moment reads

$$Z(n_s, z) = (-1)^{n_s} \prod_{p=1}^{n_s} \int_{\mathbb{R}} dr_p \frac{z}{z + e^{-r_p}} \int_C \frac{dX_{2p-1}}{4i\pi} \int_C \frac{dX_{2p}}{4i\pi} \frac{\sin(\frac{\pi}{2}(X_{2p} - X_{2p-1}))}{2\pi}$$

$$\Gamma(X_{2p-1})\Gamma(X_{2p}) e^{-\frac{r_p}{2}[X_{2p-1} + X_{2p}] + \frac{t}{24}[X_{2p-1}^3 + X_{2p}^3]} \operatorname{Pf}\left[\frac{X_i - X_j}{X_i + X_j}\right]_{2n_s \times 2n_s}. \quad (59)$$

We observe that the integrals are almost separable in $X_{2p-1}$ and $X_{2p}$ except for the sin function which couples them. Using the anti-symmetry of the Schur Pfaffian under exchange of $X_i$ and $X_j$ for any couple $(i, j)$, $i \neq j$, the addition formula

$$\sin(\frac{\pi}{2}(X_{2p} - X_{2p-1})) = \sin(\frac{\pi}{2}X_{2p})\cos(\frac{\pi}{2}X_{2p-1}) - \sin(\frac{\pi}{2}X_{2p-1})\cos(\frac{\pi}{2}X_{2p})$$

and the fact that $X_{2p}$ and $X_{2p-1}$ share the same integration measure as they share the same variable $r_p$, we rewrite the string-replicated moment as

$$Z(n_s, z) = (-1)^{n_s} \prod_{p=1}^{n_s} \int_{\mathbb{R}} dr_p \frac{z}{z + e^{-r_p}} \int_C \frac{dX_{2p-1}}{4i\pi} \int_C \frac{dX_{2p}}{4i\pi} \frac{\sin(\frac{\pi}{2}X_{2p})\cos(\frac{\pi}{2}X_{2p-1})}{\pi}$$

$$\Gamma(X_{2p-1})\Gamma(X_{2p}) e^{-\frac{r_p}{2}[X_{2p-1} + X_{2p}] + \frac{t}{24}[X_{2p-1}^3 + X_{2p}^3]} \operatorname{Pf}\left[\frac{X_i - X_j}{X_i + X_j}\right]_{2n_s \times 2n_s}. \quad (60)$$

The integrals are now separable, hence we introduce the functions

$$\phi_{2p}(X) = \frac{1}{\sqrt{\pi}}\sin(\frac{\pi}{2}X)\Gamma(X)e^{-\frac{r_p}{2}X+t\frac{X^3}{24}},$$

$$\phi_{2p-1}(X) = \frac{1}{\sqrt{\pi}}\cos(\frac{\pi}{2}X)\Gamma(X)e^{-\frac{r_p}{2}X+t\frac{X^3}{24}}.$$

(61)

Using a known property of Pfaffians (see De Bruijn [51]), we can rewrite the string-replicated moment itself as a Pfaffian

$$\prod_{\ell=1}^{2n_s}\int_C \frac{\mathrm{d}X_\ell}{4i\pi}\phi_\ell(X_\ell)\mathrm{Pf}\left[\frac{X_i-X_j}{X_i+X_j}\right]_{2n_s\times 2n_s} = \mathrm{Pf}\left[\int_C\int_C \frac{\mathrm{d}v}{4i\pi}\frac{\mathrm{d}w}{4i\pi}\phi_i(v)\phi_j(w)\frac{v-w}{v+w}\right]_{2n_s\times 2n_s}.$$

(62)

Proceeding to the rescaling $w \to 2w/t^{1/3}$ and $r \to rt^{1/3}$, the moment generating function (54) is finally given by the result announced in Section 2.1 Eqs. (5) and (6).

*Remark* 5.1. Note that in (5) we must consider matrix kernels as made up of $n_s^2$ blocks, each of which has size $2 \times 2$. Considering $2^2$ blocks of size $n_s \times n_s$ instead, would change the value of its Pfaffian by a factor $(-1)^{n_s(n_s-1)/2}$, see [12,13].

Denoting $\sigma_{t,z}(r) = \frac{z}{z+e^{-t^{1/3}r}}$, and using the definition of the Fredholm pfaffian (see e.g. Sec. 2.2. in [45] and references therein), we obtain our main new result for the case $A = +\infty$, namely an expression of the generating function as a Fredholm Pfaffian valid for any time $t$

$$\mathbb{E}_{\mathrm{KPZ}}\left[\exp\left(-ze^{H_1}\right)\right] = \mathrm{Pf}\left[J - \sigma_{t,z}K\right],$$

(63)

where the matrix kernel $K$ is given by (6), and the matrix kernel $J$ has previously been introduced in (16).

*Remark* 5.2 (Symmetry). We have the freedom to introduce an extra parameter $\beta$ so that we redefine the functions $\phi_{2p} \to \beta\phi_{2p}$ and $\phi_{2p-1} \to \frac{1}{\beta}\phi_{2p-1}$. This changes the diagonal elements $K_{11} \to \frac{1}{\beta^2}K_{11}$ and $K_{22} \to \beta^2 K_{22}$ and the off-diagonal elements remain unchanged. A possible consequence of this symmetry is that the physical relevant quantities are $K_{12}$ and the product $K_{11}K_{22}$.

## 5.2 New finite time representation as a scalar Fredholm determinant

We can use our Proposition B.2 in the Appendix to rewrite (63) as the square root of a Fredholm determinant with a scalar valued kernel.

$$\mathbb{E}_{\mathrm{KPZ}}\left[\exp\left(-ze^{H_1}\right)\right] = \sqrt{\mathrm{Det}\left[I - \bar{K}_{t,z}\right]_{\mathbb{L}^2(\mathbb{R}^+)}}.$$

(64)

The functions $f_{\mathrm{odd}}$ and $f_{\mathrm{even}}$ defined in (154) read

$$f_{\mathrm{odd}}(r) = \int_{C_v} \frac{\mathrm{d}v}{2i\pi^{3/2}}\Gamma(2v)\cos(\pi v)e^{-rv+t\frac{v^3}{3}},$$

$$f_{\mathrm{even}}(r) = \int_{C_v} \frac{\mathrm{d}v}{2i\pi^{3/2}}\Gamma(2v)\sin(\pi v)e^{-rv+t\frac{v^3}{3}}.$$

(65)

and the scalar kernel $\bar{K}_{t,z}$ is given for $x, y \geq 0$, by

$$\bar{K}_{t,z}(x,y) = 2\partial_x\int_{\mathbb{R}}\mathrm{d}r\,\frac{z}{z+e^{-r}}\left[f_{\mathrm{even}}(r+x)f_{\mathrm{odd}}(r+y)-f_{\mathrm{odd}}(r+x)f_{\mathrm{even}}(r+y)\right].$$

(66)

These forms provide an alternative formula to the one obtained at finite time in [33].

### 5.3 Long time limit of the matrix kernel

To study the long time limit we choose $z = e^{-st^{1/3}}$. Then $\sigma_{t,z}(r) \to \theta(r-s)$ where $\theta$ is the Heaviside step function. Working first on the matrix kernel of section 5.1, we obtain

$$\lim_{t \to +\infty} \mathrm{Prob}\left(\frac{H_1}{t^{1/3}} < s\right) = \mathrm{Pf}\left[J - P_s K^\infty\right]. \tag{67}$$

where $P_s$ is the projector for $r, r' \in [s, +\infty[$. Here $K^\infty$ is given by the large time limit of $K$ as follows. Starting from the definition of the $2 \times 2$ block kernel in (6), one takes the large time limit of the $\Gamma$ and trigonometric functions.

$$
\begin{aligned}
K_{11}^\infty(r, r') &= \frac{t^{1/3}}{4\pi} \int_{C_v} \int_{C_w} \frac{\mathrm{d}v\mathrm{d}w}{(2i\pi)^2} \frac{v-w}{v+w} \frac{1}{vw} e^{-rv-r'w+\frac{v^3+w^3}{3}}, \\
K_{22}^\infty(r, r') &= \frac{\pi}{4t^{1/3}} \int_{C_v} \int_{C_w} \frac{\mathrm{d}v\mathrm{d}w}{(2i\pi)^2} \frac{v-w}{v+w} e^{-rv-r'w+\frac{v^3+w^3}{3}}, \\
K_{12}^\infty(r, r') &= \frac{1}{4} \int_{C_v} \int_{C_w} \frac{\mathrm{d}v\mathrm{d}w}{(2i\pi)^2} \frac{v-w}{v+w} \frac{1}{v} e^{-rv-r'w+\frac{v^3+w^3}{3}}.
\end{aligned}
\tag{68}
$$

Using the above symmetry argument by taking $\beta = t^{1/6}/\sqrt{\pi}$, one can get rid of the time prefactors in the diagonal elements of the kernel, and we obtain that it is equivalent to the GSE kernel $K^\infty \equiv K^{\mathrm{GSE}}$ as given in Lemma 2.7. of [45]. This provides a new, independent way to show the convergence of the height distribution to the GSE at large time.

*Remark* 5.3. In Ref. [33] an alternative formula was obtained at large time, involving a (scalar) kernel $K^{\mathrm{GLD}}$ defined as

$$K^{\mathrm{GLD}}(x, y) = K_{\mathrm{Ai}}(x, y) - \frac{1}{2}\mathrm{Ai}(x) \int_0^{+\infty} \mathrm{d}z\, \mathrm{Ai}(y+z). \tag{69}$$

As we now discuss, it is possible to prove directly that the GSE Pfaffian has indeed the alternative form

$$\mathrm{Pf}[J - P_s K^\infty] = \sqrt{\mathrm{Det}(I - P_s K^{\mathrm{GLD}})}, \tag{70}$$

so that both results are consistent.

### 5.4 Long time limit of the scalar kernel

We now compute the large time limit of the scalar kernel $\bar{K}_{t,z}$ where $z = e^{-st^{1/3}}$. Rescaling the integration variables of $f_{\mathrm{odd}}$ and $f_{\mathrm{even}}$ by $t^{-1/3}$, the integration measure of $\bar{K}$ by $t^{1/3}$, one has for the auxiliary functions

$$
\begin{aligned}
f_{\mathrm{odd}}(r) &\to_{t \gg 1} f_{\mathrm{odd}}^\infty(r) = \int_{C_v} \frac{\mathrm{d}v}{4i\pi^{3/2}} \frac{1}{v} e^{-rv+\frac{v^3}{3}} = \frac{1}{2\sqrt{\pi}} \int_0^{+\infty} \mathrm{d}z \mathrm{Ai}(r+z), \\
t^{1/3} f_{\mathrm{even}}(r) &\to_{t \gg 1} f_{\mathrm{even}}^\infty(r) = \int_{C_v} \frac{\mathrm{d}v}{4i\pi^{1/2}} e^{-rv+\frac{v^3}{3}} = \frac{\sqrt{\pi}}{2} \mathrm{Ai}(r),
\end{aligned}
\tag{71}
$$

and for the scalar kernel $\bar{K}_{t,z}$

$$
\begin{aligned}
\bar{K}_{\infty,s}(x,y) &\simeq 2\partial_x \int_{\mathbb{R}} dr\, \frac{1}{1+e^{t^{1/3}(s-r)}} \left[ f_{\text{even}}^{\infty}(r+x) f_{\text{odd}}^{\infty}(r+y) - f_{\text{odd}}^{\infty}(r+x) f_{\text{even}}^{\infty}(r+y) \right] \\
&\simeq \frac{1}{2}\partial_x \int_{s}^{+\infty} dr \left[ \text{Ai}(r+x) \int_{0}^{+\infty} dz\, \text{Ai}(r+y+z) - \int_{0}^{+\infty} dz\, \text{Ai}(r+x+z)\text{Ai}(r+y) \right] \\
&\simeq \int_{s}^{+\infty} dr\, \text{Ai}(r+x)\text{Ai}(r+y) - \frac{1}{2}\text{Ai}(x+s) \int_{0}^{+\infty} dz\, \text{Ai}(y+s+z) \\
&= K_{\text{Ai}}(x+s, y+s) - \frac{1}{2}\text{Ai}(x+s) \int_{0}^{+\infty} dz\, \text{Ai}(y+s+z).
\end{aligned}
\tag{72}
$$

Hence we recover

$$
\lim_{t\to+\infty} \text{Prob}\left( \frac{H_1}{t^{1/3}} < s \right) = \sqrt{\text{Det}\left[ I - P_s K^{\text{GLD}} \right]}.
\tag{73}
$$

## 5.5 Short-time limit of the off-diagonal kernel

As required from Section 3, we now study at short time the off-diagonal element

$$
\begin{aligned}
K_{12}\left(\frac{r}{t^{\frac{1}{3}}}, \frac{r'}{t^{\frac{1}{3}}}\right) = \int_{C_v} \int_{C_w} \frac{dv\,dw}{(2i\pi)^2 \pi t^{\frac{2}{3}}} \frac{v-w}{v+w} \Gamma(2vt^{-\frac{1}{2}})\Gamma(2wt^{-\frac{1}{2}}) \\
\cos(\pi v t^{-\frac{1}{2}})\sin(\pi w t^{-\frac{1}{2}}) e^{-\frac{1}{\sqrt{t}}[rv+r'w - \frac{v^3+w^3}{3}]}.
\end{aligned}
\tag{74}
$$

We present the result obtained by a saddle point approximation applied on (74). Since the calculations are quite long, we leave the details for the Appendix C. In the short time regime, to have a non trivial correlation, we require the distance between $r$ and $r'$ to be of order $\sqrt{t}$ and we find

$$
K_{12}\left(\frac{r}{t^{\frac{1}{3}}}, \frac{r+\kappa\sqrt{t}}{t^{\frac{1}{3}}}\right) \simeq \frac{1}{2\pi\kappa t^{1/6}} \left( \sin\left(\kappa\sqrt{-W_{-1}(-\frac{t}{4}e^r)}\right) - \sin\left(\kappa\sqrt{-W_0(-\frac{t}{4}e^r)}\right) \right),
\tag{75}
$$

where $W_0$ and $W_{-1}$ are the two real branches of the Lambert function, see [47]. Taking $\kappa = 0$, we obtain the density which is positive for $r + \log(\frac{t}{4}) \leq -1$ and vanishes for $r + \log(\frac{t}{4}) = -1$ which corresponds to evaluating the Lambert functions $W(z)$ at $z = -e^{-1}$. More details about the Lambert $W$ function can be found in Appendix A.

$$
\rho(rt^{-1/3}) \simeq \frac{1}{2\pi t^{1/6}} \left( \sqrt{-W_{-1}(-\frac{t}{4}e^r)} - \sqrt{-W_0(-\frac{t}{4}e^r)} \right) \theta(r + \log(\frac{t}{4}) \leq -1).
\tag{76}
$$

We substitute $r + \log(\frac{t}{4}) \to r$ which accounts to replace $z \to \frac{t}{4}z$ as seen from the definition of $\sigma_{t,z}$ in (16). Hence, we obtain $\Xi = -1$, $\rho_\infty(a) = \frac{1}{2\pi}(\sqrt{-W_{-1}(-e^a)} - \sqrt{-W_0(-e^a)})$ and we now list useful properties of $\rho_\infty$.

1. Near the edge at $a = -1$, $\rho_\infty(a)$ vanishes as $\sqrt{-1-a}$.

2. The left asymptotics is $\rho_\infty(a) \simeq_{-a\gg1} \frac{\sqrt{-a}}{2\pi}$.

3. The right asymptotics is $\rho_\infty(a) \simeq_{a\gg1} i\frac{\sqrt{a}}{\pi}$.

### 5.6 Large deviations of the moment generating function

Taking into account the $z \to \frac{t}{4}z$ replacement, by (24) we have the large deviation principle

$$\log \mathbb{E}_{\mathrm{KPZ}}\left[\exp\left(-\frac{tze^{H_1}}{4}\right)\right] \underset{t \ll 1}{\simeq} -\frac{1}{2\pi\sqrt{t}}\int_{-\infty}^{-1}\mathrm{d}v \log\left(1+ze^v\right)\left(\sqrt{-W_{-1}(-e^v)}-\sqrt{-W_0(-e^v)}\right). \tag{77}$$

Defining $H = H_1 + \log(2\sqrt{\pi}t^{3/2})$, we obtain our first main result for the short time LDP for $A = +\infty$ in terms of the centered field $H$ (see Section 4.3 for the discussion about the centering of $H_1$)

$$\log \mathbb{E}_{\mathrm{KPZ}}\left[\exp\left(-\frac{ze^H}{8\sqrt{\pi t}}\right)\right] \underset{t \ll 1}{\simeq} -\frac{\Psi(z)}{\sqrt{t}}, \tag{78}$$

where $\Psi(z)$ is defined on $[-e, +\infty[$ as

$$\Psi(z) = \frac{1}{2\pi}\int_0^{+\infty}\mathrm{d}y\left[1-\frac{1}{y}\right]\log\left(1+zye^{-y}\right)\sqrt{y}. \tag{79}$$

To obtain (79) from (77) we performed the change of variable $y = -W(-e^v)$, so that $ye^{-y} = e^v$ and $\mathrm{d}v = \mathrm{d}y(\frac{1}{y}-1)$. Although Eq. (77) contains the two branches of the Lambert function, after the change of variable only one integral remains in (79), the branch $W_0$ indeed contributes to the range $y \in [0,1]$ and the branch $W_{-1}$ to the range $y \in [1, +\infty[$ which leads to the final range of integration $[0, +\infty[$ in (79). The minus sign in (77) disappears in the change of variable as the two branches $W_0$ and $W_{-1}$ have opposite monotonicity.

*Remark* 5.4. There is a way to express (79) in terms of a dilogarithm, defining $y = p^2$ and integrating the logarithm by part, one obtains

$$\Psi(z) = -\int_{-\infty}^{+\infty}\frac{\mathrm{d}p}{4\pi}\mathrm{Li}_2(-zp^2e^{-p^2}). \tag{80}$$

*Remark* 5.5. An integration by part on (77) leads to the expression of $f$ as

$$2\pi f(y) = \frac{2}{3}\left[-W_0(-\frac{1}{y})\right]^{3/2} - 2\left[-W_0(-\frac{1}{y})\right]^{1/2}$$
$$-\frac{2}{3}\left[-W_{-1}(-\frac{1}{y})\right]^{3/2} + 2\left[-W_{-1}(-\frac{1}{y})\right]^{1/2}. \tag{81}$$

*Remark* 5.6. It is far from obvious that on the interval $]0, e]$ the function $f$ is purely imaginary or equivalently that $\rho_\infty(a)$ is purely imaginary for $a \geq -1$. This fact is indeed true and explained in [47]. The main argument is that for $y \leq -1$, $W_0(y)$ is conjugated to $W_{-1}(y)$ in the complex sense.

The derivatives of $\Psi$ at 0 are given by $\Psi^{(q)}(0) = \frac{(-1)^{q+1}\Gamma(2q)}{2\sqrt{\pi}}q^{-\frac{3}{2}-q}4^{-q}$, allowing to determine the cumulants of $Z = e^H$, $\mathbb{E}[Z^q]^c = (-1)^{q+1}\frac{\Psi^{(q)}(0)}{\Psi'(0)^q}t^{\frac{q-1}{2}}$. We thus obtain the leading short time behavior of the cumulants of the partition sum of the directed polymer with the hard wall $A = +\infty$ as

$$\mathbb{E}\left[Z^q\right]^c = \Gamma(2q)q^{-\frac{3}{2}-q}(4\pi t)^{\frac{q-1}{2}}, \qquad q \geq 1. \tag{82}$$

One can check that for $q = 2, 3$ it exactly reproduces the results (12-13) of [33].

## 5.7 Large deviations of the distribution of $H$, $\Phi(H)$

The rate function $\Phi(H)$ for the large deviations of the distribution of $H$ is given by the solution of the optimization problem (39) over the Riemann surface of $\Psi$

$$\Phi(H) = \max_{z \geq -e} \left[ -\frac{z}{8\sqrt{\pi}} e^H + \Psi(z) \right]. \tag{83}$$

- As the left asymptotics of the density is $\rho_\infty(a) \simeq_{-a \gg 1} \frac{\sqrt{-a}}{2\pi}$, by (51) the left tail of the distribution is $\Phi(H) \simeq_{H \to -\infty} \frac{2}{15\pi} |H|^{5/2}$.

- Using (50) we obtain the cumulants: the second cumulant of $H$ is $\mathbb{E}[H^2]^c = \frac{3}{2}\sqrt{\frac{\pi t}{2}}$. It agrees with (14) in [33].

- The third cumulant of $H$ is $\mathbb{E}[H^3]^c = \left( \frac{160}{27\sqrt{3}} - \frac{27}{8} \right) \pi t$. It agrees with (15) in [33].

- The fourth cumulant of $H$ is $\mathbb{E}[H^4]^c = \frac{5}{144} \left( 567 + 486\sqrt{2} - 512\sqrt{6} \right) \pi^{3/2} t^{3/2}$.

- The fifth cumulant of $H$ is $\mathbb{E}[H^5]^c = \left( -\frac{10296145}{23328} - \frac{4725}{8\sqrt{2}} + 400\sqrt{3} + \frac{1161216}{3125\sqrt{5}} \right) \pi^2 t^2$.

- The branching field above which the continuation of $\Psi$ is required is $H_c = \log \frac{\Psi'(-e)}{\Psi'(0)} \simeq 0.9795$.

- As the right asymptotics of the density $\rho_\infty(a) \simeq_{a \gg 1} i\frac{\sqrt{a}}{\pi}$, by (52) the right tail of the distribution is $\Phi(H) \simeq_{H \to +\infty} \frac{4}{3} H^{3/2}$

- Since the density $\rho_\infty(a)$ vanishes near the edge at $a = -1$ as $\sqrt{-1-a}$, the rate function $\Phi$ is analytic.

We verify numerically that the parametric equation $\Psi'(0)e^H = \Psi'(z) - 2i\pi f'(-z)$ for $z \in [-e, 0[$ allows to obtain all $H$ in the interval $[H_c, +\infty[$ so only one continuation to $\Psi$ is required to obtain the entire rate function $\Phi$.

*Remark* 5.7. As stated in Ref. [33] Eq. (32), in the $A = +\infty$ case we have the inequality

$$\mathbb{E}_{\text{KPZ, full-space}} \left[ \exp(-z e^H) \right] < \left( \mathbb{E}_{\text{KPZ, half-space}} \left[ \exp(-z e^H) \right] \right)^2, \tag{84}$$

where both expectations are taken over the droplet IC. The inequality implies that the left tail of the full-space is at least twice the one of the half-space, which is consistent with the result obtained above.

## 6 Critical case $A = -1/2$

Recalling the definition of the field $H_1 = h(0, t) + \frac{t}{12}$, it was obtained in [35] the following Pfaffian representation for $A = -\frac{1}{2}$

$$\mathbb{E}_{\text{KPZ, 1/2 space}} \left[ \exp(-\frac{z}{4} e^{H_1}) \right] = \mathbb{E}_{\text{GOE}} \left[ \prod_{i=1}^{\infty} \frac{1}{\sqrt{1 + z e^{t^{1/3} a_i}}} \right], \tag{85}$$

where the set $\{a_i\}$ forms a Pfaffian GOE point process associated to a $2 \times 2$ matrix kernel $K$. Its off-diagonal element is defined, see Lemma 2.6 of Ref. [45], as

$$K_{12}^{\text{GOE}}(r, r') = \int_{C_v} \int_{C_w} \frac{dv\, dw}{8\pi^2} \frac{v - w}{v + w} \frac{1}{w} e^{-rv - r'w + \frac{v^3 + w^3}{3}}, \tag{86}$$

where $C_v = \varepsilon + i\mathbb{R}$ and $C_w = -\varepsilon' + i\mathbb{R}$ and $\varepsilon < \varepsilon' \in \,]0,1[$. Contrary to the other cases, here we have $\chi = \frac{1}{2}$. Note that (86) is equivalent to the formula (64) used in [36] taking into account a shift of the contour of $w$.

## 6.1 Short time limit of the off-diagonal kernel

As required from Section 3, we evaluate the off-diagonal element. Upon rescaling $(v,w) \to (v,w)t^{-1/6}$ one obtains

$$K_{12}^{\text{GOE}}(rt^{-\frac{1}{3}}, r't^{-\frac{1}{3}}) = \int_{C_v}\int_{C_w} \frac{\mathrm{d}v\mathrm{d}w}{8\pi^2 t^{\frac{1}{6}}} \frac{v-w}{v+w}\frac{1}{w} e^{-\frac{1}{\sqrt{t}}[rv+r'w-\frac{v^3+w^3}{3}]}. \tag{87}$$

We define the rate function $\varphi_r(w) = -rw + \frac{w^3}{3}$ to write the off-diagonal in a suitable form for a saddle point approximation

$$K_{12}^{\text{GOE}}(rt^{-\frac{1}{3}}, r't^{-\frac{1}{3}}) = \int_{C_v}\int_{C_w} \frac{\mathrm{d}v\mathrm{d}w}{8\pi^2 t^{\frac{1}{6}}} \frac{v-w}{v+w}\frac{1}{w} \exp\left(\frac{1}{\sqrt{t}}[\varphi_r(v) + \varphi_{r'}(w)]\right). \tag{88}$$

The saddle points are solution of $\varphi_r'(w) = 0$, i.e. $w^2 = r$ which yields two solutions

$$w^{(c)} = \pm i\sqrt{-r}, \quad \varphi(w^{(c)}) = -\frac{2}{3}w^{(c)3}, \quad \varphi''(w^{(c)}) = 2w^{(c)}. \tag{89}$$

For the saddle points to belong to the contours $C_v$ and $C_w$, we require $r \in \,]-\infty, 0]$. In the overall we have four purely imaginary saddle-point combinations indexed by $i,j = 1,2$ whose expansion yield

$$K_{12}^{\text{GOE}}(rt^{-\frac{1}{3}}, r't^{-\frac{1}{3}}) \simeq \frac{t^{1/3}}{8\pi}\sum_{i,j=1}^{2} \frac{v_i^{(c)} - w_j^{(c)}}{v_i^{(c)} + w_j^{(c)}}\frac{1}{w_j^{(c)}} \frac{\exp\left(\frac{1}{\sqrt{t}}[\varphi_r(v_i^{(c)}) + \varphi_{r'}(w_j^{(c)})]\right)}{\sqrt{-v_i^{(c)}}\sqrt{-w_j^{(c)}}}. \tag{90}$$

At the very end we are interested in the $r = r'$ limit, hence we write $r' = r + \kappa$ and aim at taking $\kappa = 0$. As $\kappa$ is small, we expand the critical point and the value of the rate function at the critical point

$$w_j^{(c)} = v_j^{(c)} + \frac{\kappa}{2v_j^{(c)}}, \qquad \varphi_{r'}(w_j^{(c)}) = \varphi_r(v_j^{(c)}) - v_j^{(c)}\kappa. \tag{91}$$

Within the linear regime, the off-diagonal kernel reads

$$K_{12}^{\text{GOE}}\left(\frac{r}{t^{\frac{1}{3}}}, \frac{r+\kappa}{t^{\frac{1}{3}}}\right) \simeq \frac{t^{1/3}}{8\pi}\sum_{i,j=1}^{2} \frac{v_i^{(c)} - v_j^{(c)} - \frac{\kappa}{2v_j^{(c)}}}{v_i^{(c)} + v_j^{(c)} + \frac{\kappa}{2v_j^{(c)}}}\frac{1}{v_j^{(c)}} \frac{\exp\left(\frac{1}{\sqrt{t}}[\varphi_r(v_i^{(c)}) + \varphi_r(v_j^{(c)}) - v_j^{(c)}\kappa]\right)}{\sqrt{-v_i^{(c)}}\sqrt{-v_j^{(c)}}}. \tag{92}$$

The leading term of this expansion is obtained for $v_i^{(c)} = -v_j^{(c)}$ as this cancels the $\varphi$ functions in the exponential and the denominator in the sum is of order $\kappa$. We additionally rescale $\kappa$ by a factor $\sqrt{t}$ to obtain

$$K_{12}^{\text{GOE}}\left(\frac{r}{t^{\frac{1}{3}}}, \frac{r+\kappa\sqrt{t}}{t^{\frac{1}{3}}}\right) \simeq \frac{1}{\pi\kappa t^{1/6}}\sin(\kappa\sqrt{-r}). \tag{93}$$

Taking $\kappa = 0$ yields the following density which vanishes at $r = 0$ and is strictly positive

$$\rho(rt^{-1/3}) \simeq \frac{1}{\pi t^{1/6}}\sqrt{-r}\,\theta(r \le 0). \tag{94}$$

Hence, we obtain $\Xi = 0$ and $\rho_\infty(a) = \frac{1}{\pi}\sqrt{-a}$ and we now list useful properties of $\rho_\infty$.

1. Near the edge at $a = 0$ $\rho_\infty(a)$ vanishes as $\sqrt{-a}$.

2. The left asymptotics is $\rho_\infty(a) \simeq_{-a \gg 1} \frac{\sqrt{-a}}{\pi}$.

3. The right asymptotics is $\rho_\infty(a) \simeq_{a \gg 1} i \frac{\sqrt{a}}{\pi}$.

## 6.2  Large deviations of the moment generating function

By (24), we obtain the Large Deviation Principle

$$\log \mathbb{E}_{\text{KPZ}}\left[\exp\left(-\frac{z}{4}e^{H_1}\right)\right] \underset{t \ll 1}{\simeq} -\frac{1}{2\pi\sqrt{t}}\int_{-\infty}^{0} \mathrm{d}v \, \log\left(1 + ze^v\right)\sqrt{-v}. \tag{95}$$

Defining $H = H_1 + \frac{1}{2}\log(\pi t)$, we obtain our main result for the short time LDP for $A = -1/2$ in terms of the centered field H

$$\log \mathbb{E}_{\text{KPZ}}\left[\exp\left(-\frac{ze^H}{\sqrt{16\pi t}}\right)\right] \underset{t \ll 1}{\simeq} -\frac{\Psi(z)}{\sqrt{t}}, \tag{96}$$

where $\Psi$ is defined on $[-1, +\infty[$ as

$$\Psi(z) = \frac{1}{2\pi}\int_{-\infty}^{0} \mathrm{d}v \, \log\left(1 + ze^v\right)\sqrt{-v} = -\frac{1}{\sqrt{16\pi}}\text{Li}_{5/2}(-z). \tag{97}$$

*Remark* 6.1. Integrating (95) by part leads to the expression of $f$ as $2\pi f(y) = \frac{2}{3}[\log y]^{3/2}$.

The derivatives of $\Psi$ at 0 are given by $\Psi^{(q)}(0) = \frac{(-1)^{q+1}\Gamma(q)}{\sqrt{16\pi}}q^{-\frac{3}{2}}$, allowing to determine the cumulants of $Z = e^H$, $\mathbb{E}[Z^q]^c = (-1)^{q+1}\frac{\Psi^{(q)}(0)}{\Psi'(0)^q}t^{\frac{q-1}{2}}$ as $\mathbb{E}[Z^q]^c = \Gamma(q)q^{-\frac{3}{2}}(16\pi t)^{\frac{q-1}{2}}$.

*Remark* 6.2. There is a way to express (97) in terms of a dilogarithm , defining $y = p^2$ and integrating the logarithm by part.

$$\Psi(z) = -\int_{-\infty}^{+\infty} \frac{\mathrm{d}p}{4\pi}\text{Li}_2(-ze^{-p^2}). \tag{98}$$

Note the resemblance between (80) and (98). A similar structure involving a dilogarithm can also be obtained for the brownian initial condition [30]. This will be further investigated in a future work.

## 6.3  Large deviations of the distribution of $H$, $\Phi(H)$

The distribution of $H$ is given by the solution of the optimization problem (39) over the Riemann surface of $\Psi$

$$\Phi(H) = \max_{z \geq -1}\left[-\frac{z}{\sqrt{16\pi}}e^H + \Psi(z)\right]. \tag{99}$$

- As the left asymptotics of the density is $\rho_\infty(a) \simeq_{-a \gg 1} \frac{\sqrt{-a}}{\pi}$ and as $\chi = \frac{1}{2}$, by (51) the left tail of the distribution is $\Phi(H) \simeq_{H \to -\infty} \frac{2}{15\pi}|H|^{5/2}$.

- The second cumulant of $H$ is $\mathbb{E}[H^2]^c = \sqrt{2\pi t}$.

- The third cumulant of $H$ is $\mathbb{E}[H^3]^c = \frac{2}{9}\left(16\sqrt{3} - 27\right)\pi t$.

- The fourth cumulant of $H$ is $\mathbb{E}[H^4]^c = \frac{8}{3}\left(18 + 15\sqrt{2} - 16\sqrt{6}\right)\pi^{3/2}t^{3/2}$.

- The fifth cumulant of $H$ is $\mathbb{E}[H^5]^c = \frac{8}{225}\left(-39625 - 27000\sqrt{2} + 36000\sqrt{3} + 6912\sqrt{5}\right)\pi^2 t^2$.

- The branching field above which the continuation of $\Psi$ is required is $H_c = \log\frac{\Psi'(-1)}{\Psi'(0)} = \log\zeta(\frac{3}{2}) \simeq 0.96026$, where $\zeta$ is the Riemann zeta function.

- As the right asymptotics of the density is $\rho_\infty(a) \simeq_{a \gg 1} i\frac{\sqrt{a}}{\pi}$, and as $\chi = \frac{1}{2}$, by (52) the right tail of the distribution is $\Phi(H) \simeq_{H \to +\infty} \frac{2}{3}H^{3/2}$.

- As the density $\rho_\infty(a)$ vanishes as $\sqrt{-a}$, the rate function $\Phi$ is analytic.

We verify numerically that the parametric equation $\Psi'(0)e^H = \Psi'(z) - 2i\pi f'(-z)$ for $z \in [-1,0[$ allows to obtain all $H$ in the interval $[H_c, +\infty[$ so only one continuation to $\Psi$ is required to obtain the entire rate function $\Phi$.

*Remark* 6.3. It turns out that the function $\Psi$ in (97) is exactly half of the large deviation function for the full space case in Ref. [28] and it leads to

$$\forall H \in \mathbb{R}, \ \Phi_{\text{half-space}}(H) = \frac{1}{2}\Phi_{\text{full-space}}(H), \tag{100}$$

where we recall that the half-space is for the critical value $A = -\frac{1}{2}$. In particular, we can use all the results derived in Ref. [28] to recover the cumulants, tails and critical points of $H$ and $P(H)$ obtained above.

Besides the tails, one can also compute the cumulants of the height using (48) and (100). We observe that $\phi_{\text{half-space}}(\frac{p}{2}) = \frac{1}{2}\phi_{\text{full-space}}(p)$ meaning that we have an explicit relation between the cumulants for the half-space and the full-space problem

$$\overline{H(t)^q}^c_{\text{half-space}} = 2^{q-1}\overline{H(t)^q}^c_{\text{full-space}}. \tag{101}$$

*Remark* 6.4. It is important to note that the coefficient of the right tail, $\frac{2}{3}H^{3/2}$, matches precisely the right tail of the GOE-TW distribution $F_1(H)$, see Ref. [52] Eqs. (1), (25) and (26), which is the *large time limit* of the critical case $A = -\frac{1}{2}$. Indeed as noted in the Remark 1.1 of [35], taking $z = e^{-st^{1/3}}$

$$\lim_{t \to \infty}\mathbb{P}\left(H(t) \leq st^{1/3}\right) = F_1(s). \tag{102}$$

This strongly suggests that the right tail is also established at short time, a fact previously noted for the full-space KPZ problem [28, 30].

# 7 Symmetric wall $A = 0$

Let us now study the case $A = 0$ for which a solution was proposed in [34]. Note that this solution is not rigorous, so our results will depend on its validity which we will assume here. Recalling the definition of the field $H_1 = h(0, t) + \frac{t}{12}$, the following Pfaffian representation for $A = 0$ was given in [34]

$$\mathbb{E}_{\text{KPZ}}\left[\exp\left(-\frac{z}{4}e^{H_1}\right)\right] = 1 + \sum_{n_s=1}^{\infty}\frac{(-1)^{n_s}}{n_s!}\prod_{p=1}^{n_s}\int_{\mathbb{R}}dr_p\frac{z}{z + e^{-t^{1/3}r_p}}\text{Pf}\left[K(r_i, r_j)\right]_{n_s \times n_s}, \tag{103}$$

where $K$ is a $2 \times 2$ block matrix with the following elements[5]

$$K_{11}(r,r') = \int_{C_v} \int_{C_w} \frac{dvdw}{16\pi^2 t^{\frac{1}{3}}} \frac{v-w}{v+w} \frac{\Gamma(\frac{1}{2}-vt^{-\frac{1}{3}})}{\Gamma(1-vt^{-\frac{1}{3}})} \frac{\Gamma(\frac{1}{2}-wt^{-\frac{1}{3}})}{\Gamma(1-wt^{-\frac{1}{3}})} e^{-rv-r'w+\frac{v^3+w^3}{3}},$$

$$K_{22}(r,r') = \int_{C_v} \int_{C_w} \frac{dvdw}{16\pi^2 t^{\frac{1}{3}}} \frac{v-w}{v+w} \frac{\Gamma(vt^{-\frac{1}{3}})}{\Gamma(\frac{1}{2}+vt^{-\frac{1}{3}})} \frac{\Gamma(wt^{-\frac{1}{3}})}{\Gamma(\frac{1}{2}+wt^{-\frac{1}{3}})} e^{-rv-r'w+\frac{v^3+w^3}{3}}, \tag{104}$$

$$K_{12}(r,r') = \int_{C_v} \int_{C_w} \frac{dvdw}{16\pi^2 t^{\frac{1}{3}}} \frac{v-w}{v+w} \frac{\Gamma(\frac{1}{2}-vt^{-\frac{1}{3}})}{\Gamma(1-vt^{-\frac{1}{3}})} \frac{\Gamma(wt^{-\frac{1}{3}})}{\Gamma(\frac{1}{2}+wt^{-\frac{1}{3}})} e^{-rv-r'w+\frac{v^3+w^3}{3}},$$

$$K_{21}(r,r') = -K_{12}(r',r).$$

The contour $C_v$ and $C_w$ must both pass at the right of 0 because of the poles as $C_{v,w} = \frac{1}{2}a_{v,w}+i\mathbb{R}$ for $a_{v,w} \in ]0,1[$ and they must be such that $v+w > 0$ for the denominators to be well defined. This representation allows can be written in a Fredholm Pfaffian form $\mathbb{E}_{\mathrm{KPZ}}\left[\exp\left(-\frac{z}{4}e^{H_1}\right)\right] = \mathrm{Pf}[J - \sigma_{t,z}K]$. As noted in [34], the large time limit of $K$ is the GSE kernel, as in the $A = +\infty$ case[6].

## 7.1 New finite time representation as a scalar Fredholm determinant

We can now use our Proposition B.2 in the Appendix to rewrite (103) as the square root of a Fredholm determinant with a scalar valued kernel.

$$\mathbb{E}_{\mathrm{KPZ}}\left[\exp\left(-\frac{z}{4}e^{H_1}\right)\right] = \sqrt{\mathrm{Det}\left[I - \bar{K}_{t,z}\right]_{\mathbb{L}^2(\mathbb{R}^+)}}. \tag{105}$$

The functions $f_{\mathrm{odd}}$ and $f_{\mathrm{even}}$ defined in (154) read

$$f_{\mathrm{odd}}(r) = \int_{C_v} \frac{dv}{4\pi} \frac{\Gamma(\frac{1}{2}-vt^{-1/3})}{\Gamma(1-vt^{-1/3})} e^{-rv+\frac{v^3}{3}},$$

$$f_{\mathrm{even}}(r) = \int_{C_v} \frac{dv}{4\pi t^{1/3}} \frac{\Gamma(vt^{-1/3})}{\Gamma(\frac{1}{2}+vt^{-1/3})} e^{-rv+\frac{v^3}{3}}. \tag{106}$$

and the scalar kernel $\bar{K}_{t,z}$ is given for $x,y \geq 0$, by

$$\bar{K}_{t,z}(x,y) = 2\partial_x \int_{\mathbb{R}} dr \frac{z}{z+e^{-t^{1/3}r}}\left[f_{\mathrm{even}}(r+x)f_{\mathrm{odd}}(r+y) - f_{\mathrm{odd}}(r+x)f_{\mathrm{even}}(r+y)\right]. \tag{107}$$

## 7.2 Long time limit of the scalar valued kernel

To study the long time limit we choose $z = e^{-st^{1/3}}$. Then $\frac{1}{1+e^{t^{1/3}(s-r)}} \to \theta(r-s)$ where $\theta$ is the Heaviside step function. Working with the scalar valued kernel (107), we show that

$$\lim_{t \to +\infty} \mathrm{Prob}\left(\frac{H_1}{t^{1/3}} < s\right) = \sqrt{\mathrm{Det}\left[I - P_s K^{\mathrm{GLD}}\right]}, \tag{108}$$

---

[5] Compared to Ref. [34], we have changed $(r,r')$ to $(-r,-r')$.

[6] The large time kernel $\tilde{K}^\infty$, given in Ref. [34], coincides with Eq. (68) of this paper, up to the change $K_{11}^\infty \to -\tilde{K}_{22}^\infty$, $K_{22}^\infty \to -\tilde{K}_{11}^\infty$ and $K_{12}^\infty \to -\tilde{K}_{21}^\infty$. This is equivalent in terms of Pfaffian since it amounts to the permutation of the columns and lines of the $2 \times 2$ block Pfaffian and an addition of a minus sign.

where $P_s$ is the projector for $r, r' \in [s, +\infty[$. In the large time regime, the auxiliary functions are given by

$$
\begin{aligned}
f_{\text{odd}}(r) \to_{t \gg 1} f_{\text{odd}}^{\infty}(r) &= \int_{C_v} \frac{dv}{4\pi^{1/2}} e^{-rv + \frac{v^3}{3}} = \frac{i\sqrt{\pi}}{2} \text{Ai}(r), \\
f_{\text{even}}(r) \to_{t \gg 1} f_{\text{even}}^{\infty}(r) &= \int_{C_v} \frac{dv}{4\pi^{3/2}} \frac{1}{v} e^{-rv + \frac{v^3}{3}} = \frac{i}{2\sqrt{\pi}} \int_0^{+\infty} dz \text{Ai}(r + z),
\end{aligned}
\tag{109}
$$

and the scalar kernel $\bar{K}_{t,z}$ converges to $\bar{K}_{\infty,s}$ given by

$$
\bar{K}_{\infty,s}(x, y) = K_{\text{Ai}}(x + s, y + s) - \frac{1}{2} \text{Ai}(x + s) \int_0^{+\infty} dz \, \text{Ai}(y + s + z). \tag{110}
$$

Hence we recover

$$
\lim_{t \to +\infty} \text{Prob}\left( \frac{H_1}{t^{1/3}} < s \right) = \sqrt{\text{Det}\left[ I - P_s K^{\text{GLD}} \right]}. \tag{111}
$$

### 7.3 Short time limit of the off-diagonal kernel

As required from Section 3, we evaluate the off-diagonal element

$$
K_{12}(rt^{-\frac{1}{3}}, r't^{-\frac{1}{3}}) = \int_{C_v} \int_{C_w} \frac{dvdw}{16\pi^2 t^{\frac{2}{3}}} \frac{v - w}{v + w} \frac{\Gamma(\frac{1}{2} - vt^{-\frac{1}{2}})}{\Gamma(1 - vt^{-\frac{1}{2}})} \frac{\Gamma(wt^{-\frac{1}{2}})}{\Gamma(\frac{1}{2} + wt^{-\frac{1}{2}})} e^{-\frac{1}{\sqrt{t}}[rv + r'w - \frac{v^3 + w^3}{3}]}. \tag{112}
$$

By Stirling's approximation and defining the rate function $\varphi_r(w) = -rw + \frac{w^3}{3}$, we write the off-diagonal in a suitable form for a saddle point approximation

$$
K_{12}(rt^{-\frac{1}{3}}, r't^{-\frac{1}{3}}) = \int_{C_v} \int_{C_w} \frac{dvdw}{16\pi^2 t^{\frac{1}{6}}} \frac{v - w}{v + w} \frac{1}{\sqrt{-v}\sqrt{w}} \exp\left( \frac{1}{\sqrt{t}} [\varphi_r(v) + \varphi_{r'}(w)] \right). \tag{113}
$$

The saddle points are solution of $\varphi_r'(w) = 0$, i.e. $w^2 = r$ which yields two solutions

$$
w^{(c)} = \pm i\sqrt{-r}, \quad \varphi(w^{(c)}) = -\frac{2}{3} w^{(c)3}, \quad \varphi''(w^{(c)}) = 2w^{(c)}. \tag{114}
$$

For the saddle points to belong to the contour $C_w$, we have the constraint $r \in ]-\infty, 0]$. In the overall we have four purely imaginary saddle-point combinations indexed by $i, j = 1, 2$ whose expansion yield

$$
K_{12}(rt^{-\frac{1}{3}}, r't^{-\frac{1}{3}}) \simeq -\frac{t^{1/3}}{16\pi} \sum_{i,j=1}^{2} \frac{v_i^{(c)} - w_j^{(c)}}{v_i^{(c)} + w_j^{(c)}} \frac{\exp\left( \frac{1}{\sqrt{t}} [\varphi_r(v_i^{(c)}) + \varphi_{r'}(w_j^{(c)})] \right)}{|w_j^{(c)}| v_i^{(c)}}. \tag{115}
$$

At the very end we are interested in the $r = r'$ limit, hence we write $r' = r + \kappa$ and aim at taking $\kappa = 0$. As $\kappa$ is small, we expand the critical point and the value of the rate function at the critical point

$$
w_j^{(c)} = v_j^{(c)} + \frac{\kappa}{2v_j^{(c)}}, \qquad \varphi_{r'}(w_j^{(c)}) = \varphi_r(v_j^{(c)}) - v_j^{(c)} \kappa. \tag{116}
$$

Within the linear regime, the off-diagonal kernel reads

$$
K_{12}\left( \frac{r}{t^{\frac{1}{3}}}, \frac{r + \kappa}{t^{\frac{1}{3}}} \right) \simeq -\frac{t^{1/3}}{16\pi} \sum_{i,j=1}^{2} \frac{v_i^{(c)} - v_j^{(c)} - \frac{\kappa}{2v_j^{(c)}}}{v_i^{(c)} + v_j^{(c)} + \frac{\kappa}{2v_j^{(c)}}} \frac{\exp\left( \frac{1}{\sqrt{t}} [\varphi_r(v_i^{(c)}) + \varphi_r(v_j^{(c)}) - v_j^{(c)} \kappa] \right)}{|v_j^{(c)}| v_i^{(c)}}. \tag{117}
$$

The leading term of this expansion is obtained for $v_i^{(c)} = -v_j^{(c)}$ as this cancels the $\varphi$ functions in the exponential and the denominator in the sum is of order $\kappa$. We additionally rescale $\kappa$ by a factor $\sqrt{t}$ to obtain

$$K_{12}\left(\frac{r}{t^{\frac{1}{3}}}, \frac{r + \kappa\sqrt{t}}{t^{\frac{1}{3}}}\right) \simeq \frac{1}{2\pi\kappa t^{1/6}} \sin(\kappa\sqrt{-r}). \tag{118}$$

Taking $\kappa = 0$ yields the following density which vanishes at $r = 0$ and is strictly positive

$$\rho(rt^{-1/3}) \simeq \frac{1}{2\pi t^{1/6}} \sqrt{-r}\,\theta(r \leq 0). \tag{119}$$

Hence, we obtain $\Xi = 0$ and $\rho_\infty(a) = \frac{1}{2\pi}\sqrt{-a}$ and we now list useful properties of $\rho_\infty$.

1. Near the edge at $a = 0$, $\rho_\infty(a)$ vanishes as $\sqrt{-a}$.

2. The left asymptotics is $\rho_\infty(a) \simeq_{-a \gg 1} \frac{\sqrt{-a}}{2\pi}$.

3. The right asymptotics is $\rho_\infty(a) \simeq_{a \gg 1} i\frac{\sqrt{a}}{2\pi}$.

### 7.4 Large deviations of the distribution of $H$

By (24), we obtain the Large Deviation Principle

$$\log \mathbb{E}_{\text{KPZ}}\left[\exp\left(-\frac{z}{4}e^{H_1}\right)\right] \underset{t \ll 1}{\simeq} -\frac{1}{2\pi\sqrt{t}} \int_{-\infty}^{0} dv \, \log\left(1 + ze^v\right)\sqrt{-v}. \tag{120}$$

Defining $H = H_1 + \frac{1}{2}\log(\pi t)$, we obtain our main result for the short time LDP for $A = 0$ in terms of the centered field H

$$\log \mathbb{E}_{\text{KPZ}}\left[\exp\left(-\frac{ze^H}{\sqrt{16\pi t}}\right)\right] \underset{t \ll 1}{\simeq} -\frac{\Psi(z)}{\sqrt{t}}, \tag{121}$$

where $\Psi$ is defined on $[-1, +\infty[$ as

$$\Psi(z) = \frac{1}{2\pi} \int_{-\infty}^{0} dv \, \log\left(1 + ze^v\right)\sqrt{-v} = -\frac{1}{\sqrt{16\pi}}\text{Li}_{5/2}(-z). \tag{122}$$

The large deviation function in (122) for $A = 0$ is strictly identical to the one in (97) for $A = -\frac{1}{2}$, therefore the distribution of the solutions for both cases will be identical, i.e.

$$\Phi_{A=0}(H) = \Phi_{A=1/2}(H) = \frac{1}{2}\Phi_{\text{full-space}}(H). \tag{123}$$

## 8 Perturbation theory of the stochastic heat equation in half-space at short time

### 8.1 Half-space SHE and its solution

We consider in this Section the perturbation theory of the half-space KPZ problem with droplet IC and Neumann b.c.

$$\forall \tau > 0, \quad \partial_x h(x, \tau)|_{x=0} = A. \tag{124}$$

Defining the partition function $Z = e^h$, we map the KPZ equation and its boundary condition to the stochastic heat equation (SHE) for $x \geq 0$

$$\partial_\tau Z(x, \tau) = \partial_x^2 Z(x, \tau) + \sqrt{2}\xi(x, \tau)Z(x, \tau), \qquad (125)$$

along with a delta IC $Z(x, 0) = \delta(x - \varepsilon)$ and Robin b.c. $\partial_x Z(x, \tau)|_{x=0} = AZ(0, \tau)$ where $\varepsilon > 0$ is introduced to regularize the solution and will be taken to $0^+$ at the end. Let $G$ be the heat kernel, i.e. $G(x, t) = \frac{1}{\sqrt{4\pi t}}\exp(-\frac{x^2}{4t})\theta(t)$ along with $G(x, 0) = \delta(x)$, then the propagator of the half-space heat equation from $(y, \tau')$ to $(x, \tau)$, see Appendix D, is

$$\mathcal{G}(y, x, \tau', \tau) = G(x - y, \tau - \tau') + G(x + y, \tau - \tau') - 2A\int_0^{+\infty} dz\, e^{-Az}G(x + y + z, \tau - \tau'). \quad (126)$$

The propagator allows us to extract the general solution of the SHE where the multiplicative noise is seen as a source term.

$$Z(x, \tau) = \int_0^\tau \int_0^{+\infty} ds dy\, \mathcal{G}(y, x, s, \tau)\big[\xi(y, s)Z(y, s) + \delta(y - \varepsilon)\delta(s)\big]$$
$$= \mathcal{G} \star (\xi Z + \delta\delta). \qquad (127)$$

We hereby define $\star$ as the space-time convolution.

## 8.2 Perturbative rescaling of the SHE at short time

Starting from the SHE (125), we choose the rescaling $\tau = t\tilde{\tau}$, $\varepsilon = \sqrt{t}\tilde{\varepsilon}$ and $x = \sqrt{t}\tilde{x}$ so that the tilde variables are of order one and the short time expansion is made in terms of powers of $t$. The equation becomes (dropping the tilde for $x$ and $\tau$)

$$\partial_\tau Z(x, \tau) = \partial_x^2 Z(x, \tau) + t^{1/4}\sqrt{2}\xi(x, \tau)Z(x, \tau) + t^{-1/2}\delta(x - \tilde{\varepsilon})\delta(\tau). \qquad (128)$$

In particular, the Robin b.c. is written as

$$\partial_x Z(x, \tau)|_{x=0} = A\sqrt{t}Z(0, \tau). \qquad (129)$$

At short time, we observe that the problem only depends on the rescaled variable $\tilde{A} = A\sqrt{t}$. As $t$ tends to zero, there are only two fixed points in this regime $\tilde{A} = +\infty$ and $\tilde{A} = 0$. It does imply the existence of two fixed points for the boundary conditions : the Dirichlet ($A = +\infty$) and the Neumann ($A$ finite) boundary conditions. A large deviation distribution will be associated to each of these boundary conditions. It explains why the half-space droplet KPZ cases $A = 0$ and $A = -\frac{1}{2}$ have the exact same large deviation distribution at short time, and it yields the generalization of the large deviation distribution to all $A$ finite.

Additionally, it has been observed in Ref. [27] using weak noise theory (WNT) that for any deterministic initial condition which is mirror-symmetric around $x = 0$, the short-time large deviation distribution of the full-space problem is twice the one of the half-space problem with the same initial condition along with the presence of a symmetric wall (i.e. $A = 0$). Using our fixed point argument, we extend this result to any finite $A$. It would be interesting to observe predictions from WNT for the other fixed point, i.e. the hard wall $A = +\infty$.

## 8.3 First two cumulants

To obtain the two first cumulants of $Z(\tilde{\varepsilon}, \tau = 1)$, we express the solution of the SHE

$$Z(\tilde{\varepsilon}, \tau = 1) = \mathcal{G} \star (\sqrt{2}t^{1/4}\xi Z + t^{-1/2}\delta\delta). \qquad (130)$$

and define successively the two first orders of the perturbation $Z_0 = t^{-1/2}\mathcal{G}\star\delta\delta$ and $Z_1 = \sqrt{2}t^{1/4}\mathcal{G}\star\xi Z_0$. In the same spirit as the perturbative expansion in Ref. [30], the leading order of the first moment is $\mathbb{E}[Z(\tilde{\varepsilon},\tau=1)] = Z_0(\varepsilon,\tau)$ and the leading order of the second moment is $\mathbb{E}[Z(\tilde{\varepsilon},\tau=1)^2]^c = \mathbb{E}\left[(Z_0+Z_1)^2(\tilde{\varepsilon},\tau=1)\right]^c = \mathbb{E}\left[Z_1(\tilde{\varepsilon},\tau=1)^2\right]$.

### 8.3.1 First moment

The zeroth order of the expansion $Z_0$ is given by the fundamental solution of the half-space heat equation as the initial condition is a Dirac.

$$Z_0(\tilde{\varepsilon},\tau=1) = \frac{1}{\sqrt{4\pi t}} + \frac{1}{\sqrt{4\pi t}}\exp(-\tilde{\varepsilon}^2) - \frac{\tilde{A}}{\sqrt{t}}e^{\tilde{A}(\tilde{A}+2\tilde{\varepsilon})}\text{Erfc}\left(\tilde{A}+\tilde{\varepsilon}\right). \qquad (131)$$

We see that the first moment is a scaling function of both variables $\tilde{A}$ and $\tilde{\varepsilon}$, they therefore are the relevant parameter to distinguish the following regimes :

1. $A$ finite and $\varepsilon = 0$ so that $\tilde{A}\ll 1$, $\tilde{\varepsilon}=0$ and $Z_0(0,t) = \frac{1}{\sqrt{\pi t}} - A + \mathcal{O}(A^2)$.

2. $A = +\infty$ and $0 < \varepsilon \ll 1$ so that $\tilde{A} = +\infty$, $\tilde{\varepsilon}$ is finite and $Z_0(\varepsilon,t) = \frac{\varepsilon^2}{\sqrt{4\pi}t^{3/2}} + \mathcal{O}(\varepsilon^4)$.

### 8.3.2 Second moment

We calculate the second moment of $Z_1$,

$$\mathbb{E}\left[Z_1(\tilde{\varepsilon},\tau=1)^2\right] = 2t^{1/2}\mathbb{E}\left[(\mathcal{G}\star\xi Z_0)(\mathcal{G}\star\xi Z_0)(\tilde{\varepsilon},\tau=1)\right].$$

Using the delta correlations of the white noise, the integrals is simplified as

$$\mathbb{E}\left[Z_1(\tilde{\varepsilon},\tau=1)^2\right] = 2t^{1/2}\int_0^1\int_0^{+\infty}\mathrm{d}s\mathrm{d}y\,\mathcal{G}(y,\tilde{\varepsilon},s,1)^2 Z_0(y,s)^2, \qquad (132)$$

and as in Section 8.3.1, we consider the two regimes at short-time

1. $\tilde{A}\ll 1$ and $\tilde{\varepsilon}=0$, up to the order 0 in $A$, $\mathbb{E}\left[Z_1(0,t)^2\right] = \sqrt{\frac{2}{\pi t}} + \mathcal{O}(A)$.

   We conclude that the rescaled second moment, up to order 0 in $A$, is

$$\frac{\mathbb{E}\left[Z(\tilde{\varepsilon},\tau=1)^2\right]}{\mathbb{E}\left[Z(\tilde{\varepsilon},\tau=1)\right]^2} = \sqrt{2\pi t} + \mathcal{O}(At). \qquad (133)$$

2. $\tilde{A} = +\infty$ and $\tilde{\varepsilon}$ finite, up to the first non zero order in $\varepsilon$, $\mathbb{E}\left[Z_1(\tilde{\varepsilon},\tau=1)^2\right] = \frac{3\varepsilon^4}{8\sqrt{2\pi}t^{5/2}}$.

   We conclude that the rescaled second moment, up to order 0 in $\varepsilon$, is

$$\frac{\mathbb{E}\left[Z(\tilde{\varepsilon},\tau=1)^2\right]}{\mathbb{E}\left[Z(\tilde{\varepsilon},\tau=1)\right]^2} = \frac{3}{2}\sqrt{\frac{\pi t}{2}}. \qquad (134)$$

## 9  Long time results

We can additionally obtain some information about the tails of the KPZ solution at large time from the limiting behavior of the off-diagonal kernel $K_{12}(r,r)$ when the variable is rescaled as $r = \tilde{r}t^{2/3}$ for large $t \gg 1$. Indeed, it was showed in [36,41], that at large time the *cumulant approximation* is still valid to obtain the far left tail of the height distribution. Indeed, defining $z = e^{-st^{1/3}}$, we have

$$\log \mathbb{P}(H(t) < st^{1/3}) \underset{t \gg 1, s < 0}{\simeq} \kappa_1, \tag{135}$$

where $\kappa_1 = -\chi \int_{\mathbb{R}} da \, \log(1 + e^{t^{1/3}(a-s)})K_{12}(a,a)$. At large time, we additionally approximate $\log(1 + e^{t^{1/3}(a-s)}) \simeq t^{1/3} \max(0, a-s)$ leading to

$$\log \mathbb{P}(H(t) < st^{1/3}) \underset{t \gg 1, s < 0}{\simeq} -t^{1/3} \chi \int_s^{+\infty} da \, (a-s)K_{12}(a,a). \tag{136}$$

This integral is dominated by the large negative argument of $K_{12}$ in the same fashion as the short-time case. Indeed, if $K_{12}(a,a) \underset{-a \gg 1}{\simeq} \beta_1 |a|^{\gamma_1}$, then

$$\log \mathbb{P}(H(t) < st^{1/3}) \underset{t \gg 1, s < 0}{\simeq} -t^{1/3} \frac{\chi \beta_1}{(\gamma_1 + 1)(\gamma_1 + 2)} |s|^{\gamma_1 + 2}. \tag{137}$$

In the large time large deviation regime where $s = \tilde{s}t^{2/3}$, with $\tilde{s}$ fixed, this expression takes the form $\log \mathbb{P}(H(t) < \tilde{s}t) \underset{t \gg 1, s < 0}{\simeq} -t^{\frac{5+2\gamma_1}{3}} \frac{\chi \beta_1}{(\gamma_1 + 1)(\gamma_1 + 2)} |\tilde{s}|^{\gamma_1 + 2}$. In all observed cases, we have $\gamma_1 = \frac{1}{2}$. For the rest of this section, we apply (137) to the half-space droplet KPZ case with $A = -\frac{1}{2}, 0, +\infty$ and argue that for all cases, we have

$$\log \mathbb{P}(H(t) < \tilde{s}t) \underset{t \gg 1, \tilde{s} < 0}{\simeq} -t^2 \frac{2}{15\pi} |\tilde{s}|^{5/2}. \tag{138}$$

As the cases $A = +\infty$ and $A = 0$ share the same kernel at large time, i.e. the GSE kernel, we will study them together.

### 9.1  $A = +\infty$ and $A = 0$

It has been shown in the Appendix of [36] that the large negative behavior of the GOE kernel is the edge of Wigner's semi-circle $K_{12}^{\text{GOE}}(a,a) \underset{-a \gg 1}{\simeq} \frac{\sqrt{|a|}}{\pi}$. The off-diagonal GSE kernel (68) differs from the off-diagonal GOE kernel (86) by two aspects

- There is an extra factor $\frac{1}{2}$ in the GSE kernel.

- In the GSE kernel, both integrals in the definition have their contour at the right of 0 while in the GOE kernel, one contour is on the left and the other one is on the right.

In the large deviation regime, the position of the contour with respect to zero is without importance as both (68) and (86) are evaluated through a saddle point method where the saddle points is located on the imaginary axis. This implies that

$$K_{12}^{\text{GSE}}(a,a) \underset{-a \gg 1}{\simeq} \frac{\sqrt{|r|}}{2\pi} \theta(-r). \tag{139}$$

Therefore, by (137), we obtain the left tail of the distribution

$$\log \mathbb{P}(H(t) < \tilde{s}t) \underset{t \gg 1, s < 0}{\simeq} -t^2 \frac{2}{15\pi} |\tilde{s}|^{5/2}. \tag{140}$$

## 9.2 $A = -\frac{1}{2}$

For the critical case for droplet initial condition, we have obtained in [41] the full large deviation rate function for the left tail which describes the crossover between the 5/2 tail and the cubic tail of GOE Tracy-Widom. Indeed at large time, we have the large deviation principle for $\tilde{s} < 0$

$$-\lim_{t\to\infty} \frac{1}{t^2} \log \mathbb{P}(H(t) < \tilde{s}t) = \Phi_-^{\text{half-space}}(\tilde{s}) = \frac{1}{2}\Phi_-^{\text{full-space}}(\tilde{s}), \tag{141}$$

where $\Phi_-^{\text{full-space}}(\tilde{s})$ was obtained explicitly in Refs. [39, 41]. The limiting behavior of $\Phi_-^{\text{half-space}}$ are

$$\Phi_-^{\text{half-space}}(\tilde{s}) \simeq_{-\tilde{s}\ll 1} \frac{1}{24}|\tilde{s}|^3, \qquad \Phi_-^{\text{half-space}}(\tilde{s}) \simeq_{-\tilde{s}\gg 1} \frac{2}{15\pi}|\tilde{s}|^{5/2}. \tag{142}$$

The small $\tilde{s}$ behavior indeed matches the GOE Tracy-Widom behavior given in Table 1.

### 9.3 Further conjecture

In view of the coefficients obtained for the two limiting behavior of $\Phi_-^{\text{half-space}}(\tilde{s})$ for $A = 0$ and $A = \infty$ in Eq. (140) ($H^{5/2}$ tail ) and in Table 1 ($H^3$ tail), it is tempting to conjecture that the full large deviation rate function for the left tail does not depend on $A$ for $A > -\frac{1}{2}$. Note that this is also consistent with the bound of Eq. (84).

## 10 Conclusion

We have developed a mathematical framework enabling to easily derive the short time properties of the large deviations of the distribution of the height of the KPZ solutions. Notably, when there exists a Pfaffian representation for the moment generating function of the KPZ solution, the short time large deviation rate function only depends on the asymptotics density of the associated Pfaffian point process. We have exploited this method to study KPZ in a half-space with three different boundary conditions $A = 0, -1/2, +\infty$.

Furthermore, we have obtained a new Pfaffian representation of the KPZ solution for the hard wall boundary condition which is valid at all times and also allows to study easily the short time properties. We have additionally extended the *cumulant approximation* that was previously introduced in [36, 41] to obtain the left tail of the distribution at large time. This approximation allows in the short-time context to obtain the entire height distribution of the KPZ solution. It is quite remarkable that this method, i.e. the truncation to the first cumulant, also contains all the information in the short time limit.

On a more technical side, we have obtained a general method to transform a class of Fredholm Pfaffians with a $2 \times 2$ block kernel into a Fredholm determinants with a scalar valued kernel. This extends some of the results of Refs. [53, 54].

We hope that this effort will motivate further bridges with different theoretical methods such as Weak Noise Theory or with numerical simulations.

## Acknowledgements

We acknowledge motivating discussions with Ivan Corwin, Promit Ghosal, Baruch Meerson, Sylvain Prolhac, Gregory Schehr, Satya N. Majumdar and Li-Cheng Tsai. We particularly thank Guillaume Barraquand regarding discussions about the equivalence between Pfaffian and determinantal representations. We finally acknowledge support from ANR grant ANR-17-CE30-0027-01 RaMaTraF.

## A   The Lambert function $W$

We introduce the Lambert $W$ function [47] which we use throughout this paper to study the solution with droplet initial condition and hard wall boundary condition. Consider the function defined on $\mathbb{C}$ by $f(z) = ze^z$, the $W$ function is composed of all inverse branches of $f$ so that $W(ze^z) = z$. It does have two real branches, $W_0$ and $W_{-1}$ defined respectively on $[-e^{-1}, +\infty[$ and $[-e^{-1}, 0[$. On their respective domains, $W_0$ is strictly increasing and $W_{-1}$ is strictly decreasing. By differentiation of $W(z)e^{W(z)} = z$, one obtains a differential equation valid for all branches of $W(z)$

$$\frac{dW}{dz}(z) = \frac{W(z)}{z(1 + W(z))}. \tag{143}$$

Concerning their asymptotics, $W_0$ behaves logarithmically for large argument $W_0(z) \simeq_{z \to +\infty} \ln(z) - \ln\ln(z)$ and is linear for small argument $W_0(z) \simeq_{z \to 0} z - z^2 + \mathcal{O}(z^3)$. $W_{-1}$ behaves logarithmically for small argument $W_{-1}(z) \simeq_{z \to 0^-} \ln(-z) - \ln(-\ln(-z))$. Both branches join smoothly at the point $z = -e^{-1}$ and have the value $W(-e^{-1}) = -1$. These remarks are summarized on Fig. 2. More details on the other branches, $W_k$ for integer $k$, can be found in [47].

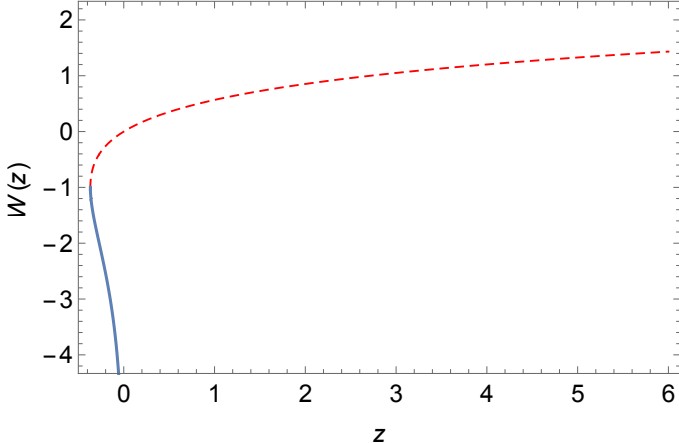

Figure 2: The Lambert function $W$. The dashed red line corresponds to the branch $W_0$ whereas the blue line corresponds to the branch $W_{-1}$.

# B  Representation of a $2 \times 2$ block Fredholm Pfaffian as a Fredholm determinant

In this Appendix, we present a new representation of a class of $2 \times 2$ block Fredholm Pfaffian with a matrix valued kernel in terms of a Fredholm determinant with a scalar valued kernel. This Appendix is an extension to arbitrary time of the arguments presented by G. Barraquand [55] for the case of the GSE kernel where the associated scalar valued kernel is the one found in [33].

Consider a measure $\mu$ on a contour $C \in \mathbb{C}$ and another measure $\nu_z$ on the real line $\mathbb{R}$, depending on a real parameter $z$. Consider the quantity $Q(z)$ defined by

$$Q(z) = 1 + \sum_{n_s=1}^{\infty} \frac{(-1)^{n_s}}{n_s!} Z(n_s, z) \tag{144}$$

and

$$Z(n_s, z) = \prod_{p=1}^{n_s} \int_{\mathbb{R}} \nu_z(\mathrm{d}r_p) \int_C \mu(\mathrm{d}X_{2p-1}) \int_C \mu(\mathrm{d}X_{2p})$$

$$\phi_{\mathrm{odd}}(X_{2p-1})\phi_{\mathrm{even}}(X_{2p})e^{-r_p[X_{2p-1}+X_{2p}]} \mathrm{Pf}\left[\frac{X_i - X_j}{X_i + X_j}\right]_{i,j=1}^{2n_s}. \tag{145}$$

Then we have the two following lemma and proposition.

**Lemma B.1.** $Q(z)$ is equal to a Fredholm Pfaffian with a $2 \times 2$ matrix valued skew-symmetric kernel

$$Q(z) = \mathrm{Pf}[J - K]_{\mathbb{L}^2(\mathbb{R}, \nu_z)}. \tag{146}$$

For $r, r' \in \mathbb{R}$ the matrix kernel $K$ is given by

$$K_{11}(r, r') = \int_C \int_C \mu(\mathrm{d}v)\mu(\mathrm{d}w)\frac{v - w}{v + w}\phi_{\mathrm{odd}}(v)\phi_{\mathrm{odd}}(w)e^{-rv-r'w},$$

$$K_{22}(r, r') = \int_C \int_C \mu(\mathrm{d}v)\mu(\mathrm{d}w)\frac{v - w}{v + w}\phi_{\mathrm{even}}(v)\phi_{\mathrm{even}}(w)e^{-rv-r'w},$$

$$K_{12}(r, r') = \int_C \int_C \mu(\mathrm{d}v)\mu(\mathrm{d}w)\frac{v - w}{v + w}\phi_{\mathrm{odd}}(v)\phi_{\mathrm{even}}(w)e^{-rv-r'w}, \tag{147}$$

$$K_{21}(r, r') = \int_C \int_C \mu(\mathrm{d}v)\mu(\mathrm{d}w)\frac{v - w}{v + w}\phi_{\mathrm{even}}(v)\phi_{\mathrm{odd}}(w)e^{-rv-r'w}.$$

and the matrix kernel $J$ is defined by

$$J(r, r') = \begin{pmatrix} 0 & 1 \\ -1 & 0 \end{pmatrix}\mathbb{1}_{r=r'}. \tag{148}$$

*Proof.* Using a known property of Pfaffians (see De Bruijn [51]),

$$\prod_{p=1}^{2n_s} \int_C \mu(\mathrm{d}X_p)\,\tilde{\phi}_p(X_p)\mathrm{Pf}\left[\frac{X_i - X_j}{X_i + X_j}\right]_{2n_s \times 2n_s} =$$

$$\mathrm{Pf}\left[\int_C \int_C \mu(\mathrm{d}v)\mu(\mathrm{d}w)\,\tilde{\phi}_i(v)\tilde{\phi}_j(w)\frac{v - w}{v + w}\right]_{2n_s \times 2n_s} \tag{149}$$

and specifying the functions $\tilde{\phi}$ as

$$\tilde{\phi}_{2p-1}(X) = \phi_{\text{odd}}(X)e^{-r_p X}, \quad \tilde{\phi}_{2p}(X) = \phi_{\text{even}}(X)e^{-r_p X}, \tag{150}$$

we rewrite $Z(n_s, z)$ as

$$Z(n_s, z) = \prod_{p=1}^{n_s} \int_{\mathbb{R}} \nu_z(\mathrm{d}r_p) \mathrm{Pf}\big[K(r_i, r_j)\big]_{i,j=1}^{n_s}. \tag{151}$$

Noticing that each couple $(r_i, r_j)$ appears in $2 \times 2$ block, the definition of the matrix valued kernel $K$ in (147) follows. Note that in (151) we must consider matrix kernels as made up of $n_s^2$ blocks, each of which has size $2 \times 2$. Considering $2^2$ blocks of size $n_s \times n_s$ instead, would change the value of its Pfaffian by a factor $(-1)^{n_s(n_s-1)/2}$, see [12,13]. Note that $K_{21}(r, r') = -K_{12}(r', r)$ so the kernel is skew-symmetric. Finally, from [42] Section 8, $Q(z)$ is by definition a Fredholm Pfaffian

$$Q(z) = \mathrm{Pf}[J - K]_{\mathbb{L}^2(\mathbb{R}, \nu_z)}. \tag{152}$$

$\square$

**Proposition B.2.** *$Q(z)$ is equal to the square root of a Fredholm determinant with scalar valued kernel*

$$Q(z) = \sqrt{\mathrm{Det}\big[I - \bar{K}\big]_{\mathbb{L}^2(\mathbb{R}^+)}}, \tag{153}$$

*where $\mathbb{L}^2(\mathbb{R}^+)$ is considered with the uniform measure. Introducing the functions defined on $\mathbb{R}$ by*

$$f_{\text{odd}}(r) = \int_C \mu(\mathrm{d}\nu)\,\phi_{\text{odd}}(\nu)e^{-r\nu}, \quad f_{\text{even}}(r) = \int_C \mu(\mathrm{d}\nu)\,\phi_{\text{even}}(\nu)e^{-r\nu}, \tag{154}$$

*the scalar kernel $\bar{K}$ is given, for $x, y \in \mathbb{R}^+$, by*

$$\bar{K}(x, y) = 2\int_{\mathbb{R}} \nu_z(\mathrm{d}r)\left[f'_{\text{even}}(r+x)f_{\text{odd}}(r+y) - f'_{\text{odd}}(r+x)f_{\text{even}}(r+y)\right] \tag{155}$$

*and the scalar kernel $I$ is the identity kernel $I(x, y) = \mathbb{1}_{x=y}$.*

*Proof.* We start back from the definition of the matrix valued kernel $K$ in Eq. (147) and use the following identity

$$\frac{\nu - w}{\nu + w} = \frac{w}{w} - \frac{2w}{\nu + w}, \tag{156}$$

along with the identities valid for $\mathrm{Re}(w) > 0$ and $\mathrm{Re}(\nu + w) > 0$

$$\frac{1}{w} = \int_0^{+\infty} \mathrm{d}x\, e^{-xw}, \quad \frac{1}{\nu + w} = \int_0^{+\infty} \mathrm{d}x\, e^{-x(\nu+w)}, \quad we^{-r'w} = -\partial_{r'}e^{-r'w}. \tag{157}$$

These identities are used to separate the integrals w.r.t the variables $\nu$ and $w$. One can now introduce the odd and even functions

$$f_{\text{odd}}(r) = \int_C \mu(\mathrm{d}\nu)\,\phi_{\text{odd}}(\nu)e^{-r\nu}, \qquad f_{\text{even}}(r) = \int_C \mu(\mathrm{d}\nu)\,\phi_{\text{even}}(\nu)e^{-r\nu}, \tag{158}$$

to write the elements of the kernel $K$ as

$$K_{11}(r,r') = \int_0^{+\infty} dx \left[ 2f_{odd}(r+x)f'_{odd}(r'+x) - f_{odd}(r)f'_{odd}(r'+x) \right],$$

$$K_{22}(r,r') = \int_0^{+\infty} dx \left[ 2f_{even}(r+x)f'_{even}(r'+x) - f_{even}(r)f'_{even}(r'+x) \right],$$

$$K_{12}(r,r') = \int_0^{+\infty} dx \left[ 2f_{odd}(r+x)f'_{even}(r'+x) - f_{odd}(r)f'_{even}(r'+x) \right], \tag{159}$$

$$K_{21}(r,r') = \int_0^{+\infty} dx \left[ 2f_{even}(r+x)f'_{odd}(r'+x) - f_{even}(r)f'_{odd}(r'+x) \right].$$

Consider the notation for the matrix valued kernel

$$K(r,r') = \begin{pmatrix} K_{11}(r,r') & K_{12}(r,r') \\ K_{21}(r,r') & K_{22}(r,r') \end{pmatrix}. \tag{160}$$

One of the two main steps of the proof is to notice that the kernel $K$ can be factorized as a product of a matrix that depends only on $r$ and another matrix that depends only on $r'$.

$$K(r,r') = \int_0^{+\infty} dx \begin{pmatrix} 2f_{odd}(r+x) - f_{odd}(r) \\ 2f_{even}(r+x) - f_{even}(r) \end{pmatrix} \begin{pmatrix} f'_{odd}(r'+x) \\ f'_{even}(r'+x) \end{pmatrix}^T. \tag{161}$$

We now write this matrix product as an operator product

$$K(r,r') = \int_0^{+\infty} dx\, A^{(1)}(r,x)A^{(2)}(x,r'), \tag{162}$$

where the Hilbert-Schmidt operator $A^{(1)} : \mathbb{L}^2(\mathbb{R}, \nu_z) \to \mathbb{L}^2(\mathbb{R}^+)$ is defined by

$$A^{(1)}(r,x) = \begin{pmatrix} 2f_{odd}(r+x) - f_{odd}(r) \\ 2f_{even}(r+x) - f_{even}(r) \end{pmatrix}. \tag{163}$$

and $A^{(2)} : \mathbb{L}^2(\mathbb{R}^+) \to \mathbb{L}^2(\mathbb{R}, \nu_z)$ is defined by

$$A^{(2)}(x,r') = \begin{pmatrix} f'_{odd}(r'+x) \\ f'_{even}(r'+x) \end{pmatrix}^T. \tag{164}$$

We have,

$$\mathrm{Pf}[J-K]_{\mathbb{L}^2(\mathbb{R}, \nu_z)} = \mathrm{Pf}\left[J - A^{(1)}A^{(2)}\right]_{\mathbb{L}^2(\mathbb{R}, \nu_z)}. \tag{165}$$

Using that for a skew-symmetric kernel $K$, $\mathrm{Pf}[J-K]^2 = \mathrm{Det}[I+JK]$, (see [42], Lemma 8.1), where the scalar kernel $I$ is the identity kernel $I(x,y) = \mathbb{1}_{x=y}$, this gives

$$\mathrm{Pf}[J-K]^2_{\mathbb{L}^2(\mathbb{R}, \nu_z)} = \mathrm{Det}\left[I + JA^{(1)}A^{(2)}\right]_{\mathbb{L}^2(\mathbb{R}, \nu_z)}. \tag{166}$$

Following [53, 56], one uses the "needlessly fancy" general relation $\det(I+AB) = \det(I+BA)$ for arbitrary Hilbert-Schmidt operators $A$ and $B$. They may act between different spaces as long as the products make sense. In the present context $\det(I+AB)$ is the Fredholm determinant of a matrix valued kernel whilst $\det(I+BA)$ is a Fredholm determinant of scalar valued kernel.

$$\mathrm{Pf}[J-K]^2_{\mathbb{L}^2(\mathbb{R}, \nu_z)} = \mathrm{Det}\left[I + A^{(2)}JA^{(1)}\right]_{\mathbb{L}^2(\mathbb{R}^+)}. \tag{167}$$

Let us compute the scalar valued kernel of the operator $A^{(2)}JA^{(1)} : \mathbb{L}^2(\mathbb{R}^+) \to \mathbb{L}^2(\mathbb{R}^+)$.

$$A^{(2)}JA^{(1)}(x,y) = \int_{\mathbb{R}} \nu_z(\mathrm{d}r) A^{(2)}(x,r)JA^{(1)}(r,y) = \mathcal{K}(x,y) - \frac{1}{2}\mathcal{K}(x,0), \tag{168}$$

where $\mathcal{K} : \mathbb{L}^2(\mathbb{R}^+) \to \mathbb{L}^2(\mathbb{R}^+)$ is defined by

$$\mathcal{K}(x,y) = 2 \int_{\mathbb{R}} \nu_z(\mathrm{d}r) \left[ f'_{\mathrm{odd}}(r+x) f_{\mathrm{even}}(r+y) - f'_{\mathrm{even}}(r+x) f_{\mathrm{odd}}(r+y) \right]. \tag{169}$$

We observe that $\mathcal{K}$ can be written as a partial derivative w.r.t its first variable $\mathcal{K}(x,y) = \partial_x k(x,y)$ where $k$ is a skew-symmetric scalar kernel given by

$$k(x,y) = 2 \int_{\mathbb{R}} \nu_z(\mathrm{d}r) \left[ f_{\mathrm{odd}}(r+x) f_{\mathrm{even}}(r+y) - f_{\mathrm{even}}(r+x) f_{\mathrm{odd}}(r+y) \right]. \tag{170}$$

The operator $(x,y) \mapsto \mathcal{K}(x,0)$ is of rank 1 and can be written as $\mathcal{K} |\delta\rangle \langle 1|$ where all products have to be taken in the sense of Hilbert-Schmidt integral operator products. Here $\delta$ is the $\delta$-function at $x = 0$ and 1 denotes the function $1(x) = 1$ for all $x \geq 0$. This leads to the equality

$$\mathrm{Pf}[J-K]^2_{\mathbb{L}^2(\mathbb{R},\nu_z)} = \mathrm{Det}\left[ I + \mathcal{K}\left( I - \frac{1}{2}|\delta\rangle\langle 1| \right) \right]_{\mathbb{L}^2(\mathbb{R}^+)}. \tag{171}$$

As $|\delta\rangle\langle 1|$ is of rank 1, by the matrix determinant lemma, we have

$$\mathrm{Pf}[J-K]^2_{\mathbb{L}^2(\mathbb{R},\nu_z)} = \mathrm{Det}[I+\mathcal{K}]_{\mathbb{L}^2(\mathbb{R}^+)}\left( 1 - \frac{1}{2}\langle 1|\mathcal{K}(I+\mathcal{K})^{-1}|\delta\rangle \right). \tag{172}$$

We now want to prove the following identity to be able to conclude

$$\langle 1|\mathcal{K}(I+\mathcal{K})^{-1}|\delta\rangle = 0. \tag{173}$$

The main ingredient to prove this is the remarkable fact that $\mathcal{K}$ is expressed as a product $\mathcal{K} = Dk$, where $D = \partial_x$ and $k$ is a skew-symmetric kernel as introduced in (170). For this type of kernels, (173) was proven in [12,13] Appendix H, and we re-derive the proof here for completeness. We first expand $(I+\mathcal{K})^{-1}$ as a series

$$\langle 1|\mathcal{K}(I+\mathcal{K})^{-1}|\delta\rangle = -\sum_{n=1}^{+\infty}(-1)^n \langle 1|\mathcal{K}^n|\delta\rangle. \tag{174}$$

A sufficient condition for (173) to hold is that for all $n \geq 1$, $\langle 1|\mathcal{K}^n|\delta\rangle = 0$. We introduce the notation $Q_n = \langle 1|\mathcal{K}^n|\delta\rangle$, show explicitly for $n=1$ that this is true and we will finally proceed by induction.

$$Q_1 = \int_{\mathbb{R}^{+,\otimes 2}} \mathrm{d}x_1 \mathrm{d}x_2\, \mathcal{K}(x_1,x_2)\delta(x_2) = \int_{\mathbb{R}^+} \mathrm{d}x_1\, \partial_{x_1}k(x_1,0) = 0, \tag{175}$$

where we used that $k(0,0) = 0$. The general term $Q_n$ is given by

$$\begin{aligned}
Q_n &= \int_{\mathbb{R}^{+,\otimes(n+1)}} \mathrm{d}x_1 \dots \mathrm{d}x_{n+1}\, \mathcal{K}(x_1,x_2)\prod_{i=2}^{n}\left[\mathcal{K}(x_i,x_{i+1})\right]\delta(x_{n+1}) \\
&= \int_{\mathbb{R}^{+,\otimes(n-1)}} \mathrm{d}x_2 \dots \mathrm{d}x_n\, k(x_2,0)\prod_{i=2}^{n-1}\left[\partial_{x_i}k(x_i,x_{i+1})\right]\partial_{x_n}k(x_n,0),
\end{aligned} \tag{176}$$

where in the last line, we have integrated w.r.t $x_1$ and $x_{n+1}$. We shall prove that $Q_n$ verifies the following identity for any $n \geq 2$

$$Q_n = \frac{1}{2} \sum_{p=1}^{n-1} Q_p Q_{n-p}. \tag{177}$$

To do so, we use two observations. Due to the skew-symmetry of $k$, for any $x$, we have $\partial_x k(x, y) = -\partial_x k(y, x)$. Besides, all derivatives are applied on the first variable of $k$ in the definition of $Q_n$, so we use the integration by part identity (coupled to the skew-symmetry of $k$)

$$\int_y \partial_x k(x, y) \partial_y k(y, z) = \partial_x k(x, 0) k(z, 0) + \int_y \partial_x \partial_y k(y, x) k(y, z). \tag{178}$$

The idea is to push all derivatives from the right to the left in the second line of (176) by successive integrations by part. The boundary terms in (178) will cut the integral into two parts, giving the discrete convolution term $Q_p Q_{n-p}$. The very last term not coming from the boundaries will be

$$\int_{\mathbb{R}^{+,\otimes(n-1)}} dx \, \partial_{x_2} k(0, x_2) \prod_{i=2}^{n-1} \left[ \partial_{x_{i+1}} k(x_{i+1}, x_i) \right] k(x_n, 0). \tag{179}$$

Up to the relabeling $x_i \to x_{n+1-i}$ one recognizes that this is equal to $-Q_n$ leading to

$$Q_n = -Q_n + \sum_{p=1}^{n-1} Q_p Q_{n-p}, \tag{180}$$

which is exactly (177). Finally, from (177) and the fact that $Q_1 = 0$, by induction we have for all $n \geq 1$, $Q_n = 0$. And therefore (173) is verified so that the Fredholm Pfaffian reads

$$\mathrm{Pf}[J - K]_{\mathbb{L}^2(\mathbb{R}, \nu_z)}^2 = \mathrm{Det}[I + \mathcal{K}]_{\mathbb{L}^2(\mathbb{R}^+)}. \tag{181}$$

Defining $\bar{K} = -\mathcal{K}$ and taking the square root on both side of (181) ends the proof. $\qquad \square$

*Remark* B.1. By expanding on powers of $\tau$, one can check that

$$\mathrm{Det}\left[ I + \tau \int_0^{+\infty} dx \begin{pmatrix} g_1(r, x) \\ g_2(r, x) \end{pmatrix} \begin{pmatrix} g_3(x, r') & g_4(x, r') \end{pmatrix} \right]_{\mathbb{L}^2(\mathbb{R}, \nu_z)}$$
$$= \tag{182}$$
$$\mathrm{Det}\left[ I + \tau \int_{\mathbb{R}} \nu_z(dr) \begin{pmatrix} g_3(x, r) & g_4(x, r) \end{pmatrix} \begin{pmatrix} g_1(r, y) \\ g_2(r, y) \end{pmatrix} \right]_{\mathbb{L}^2(\mathbb{R}^+)}.$$

It is easy to see that the traces of arbitrary powers of the two operators of (182) coincide.

*Remark* B.2. A shorter version of the proof concerning the identity $\langle 1 | \mathcal{K}(I + \mathcal{K})^{-1} | \delta \rangle = 0$ can be given as follows. Since $\mathcal{K} = Dk$, we rewrite this identity as

$$Q = \langle 1 | \mathcal{K}(I + \mathcal{K})^{-1} | \delta \rangle = -\langle \delta | k(I + Dk)^{-1} | \delta \rangle. \tag{183}$$

where we used for any function $f$ that $\langle 1 | Df = -\langle \delta | f$. Note also the commutation relation $k(I + Dk)^{-1} = (I + kD)^{-1} k$. We recall that $D$ is the derivative operator defined by its matrix element $\langle f | D | g \rangle = \int_{\mathbb{R}^+} dx \, f(x) g'(x)$. By integration by part, the adjoint of $D$ is

$$D^T = -D - |\delta\rangle\langle\delta|. \tag{184}$$

Taking the adjoint of the operator $k(I + Dk)^{-1}$, we have

$$Q = -\langle\delta|(I + k^T D^T)^{-1}k^T|\delta\rangle = \langle\delta|\left(I + kD + k|\delta\rangle\langle\delta|\right)^{-1}k|\delta\rangle. \tag{185}$$

We can use the Sherman-Morrison identity since the last term in the inverse is a rank 1 operator.

$$Q = \langle\delta|(I + kD)^{-1}k|\delta\rangle - \frac{\langle\delta|(I + kD)^{-1}k|\delta\rangle\langle\delta|(I + kD)^{-1}k|\delta\rangle}{1 + \langle\delta|(I + kD)^{-1}k|\delta\rangle} = -Q - \frac{Q^2}{1 - Q}, \tag{186}$$

which implies $Q = 0$ or $Q = 2$. Since the amplitude of $k$ can be increased continuously from 0 to any value, by continuity, the solution is $Q = 0$. This agrees with the previous calculation using a power series expansion.

*Remark* B.3. As these proofs did not depend on the measures $\nu$ and $\mu$, we can apply these lemmas to the solution of the KPZ equation at all times.

# C  Short time limit of the $A = \infty$ kernel

As required from Section 3, we study at short time the off-diagonal element

$$K_{12}(rt^{-\frac{1}{3}}, r't^{-\frac{1}{3}}) = \int_{C_\nu}\int_{C_w} \frac{d\nu dw}{(2i\pi)^2\pi t^{\frac{2}{3}}}\frac{\nu - w}{\nu + w}\Gamma(2\nu t^{-\frac{1}{2}})\Gamma(2wt^{-\frac{1}{2}})$$
$$\cos(\pi\nu t^{-\frac{1}{2}})\sin(\pi w t^{-\frac{1}{2}})e^{-\frac{1}{\sqrt{t}}[r\nu + r'w - \frac{\nu^3 + w^3}{3}]}. \tag{187}$$

We rewrite the pre-factors of the exponential in the integrand of $K_{12}$ using the following continuations in the complex plane and their associated Stirling asymptotics

$$\Gamma(2w)\sin(\pi w) = \frac{2^{2w}\sqrt{\pi}}{2}\frac{\Gamma(w + \frac{1}{2})}{\Gamma(1 - w)} \underset{|w|\gg 1}{\simeq} \sqrt{\frac{-\pi}{4w}}\exp\left((\log(-4w^2) - 2)w\right),$$
$$\Gamma(2\nu)\cos(\pi\nu) = \frac{2^{2\nu}\sqrt{\pi}}{2}\frac{\Gamma(\nu)}{\Gamma(\frac{1}{2} - \nu)} \underset{|\nu|\gg 1}{\simeq} \sqrt{\frac{\pi}{4\nu}}\exp\left((\log(-4\nu^2) - 2)\nu\right). \tag{188}$$

Within these approximations and continuations, the off-diagonal kernel reads

$$K_{12}(rt^{-\frac{1}{3}}, r't^{-\frac{1}{3}}) = \int_{C_\nu}\int_{C_w} \frac{d\nu dw}{4(2i\pi)^2 t^{\frac{1}{6}}}\frac{\nu - w}{\nu + w}\frac{1}{\sqrt{-w}}\frac{1}{\sqrt{\nu}}$$
$$\exp\left(-\frac{1}{\sqrt{t}}[(r + 2 - \log(\frac{-4\nu^2}{t}))\nu + (r' + 2 - \log(\frac{-4w^2}{t}))w - \frac{\nu^3 + w^3}{3}]\right). \tag{189}$$

We define the rate function $\varphi_r(w) = (-r + \log(\frac{-4w^2}{t}) - 2)w + \frac{w^3}{3}$ to write the off-diagonal in a suitable form for a saddle point approximation

$$K_{12}(rt^{-\frac{1}{3}}, r't^{-\frac{1}{3}}) = \int_{C_\nu}\int_{C_w} \frac{d\nu dw}{4(2i\pi)^2 t^{\frac{1}{6}}}\frac{\nu - w}{\nu + w}\frac{1}{\sqrt{-w}}\frac{1}{\sqrt{\nu}}\exp\left(\frac{1}{\sqrt{t}}[\varphi_r(\nu) + \varphi_{r'}(w)]\right). \tag{190}$$

The saddle points are solution of $\varphi'_r(w) = 0$, i.e. $w^2 + \log(-w^2) = r + \log(\frac{t}{4})$ which yields four solutions expressed with the two real branches of the Lambert function $W_{0,-1}$, see [47]

$$w^{(c)} = \pm i\sqrt{-W_{0,-1}(-\frac{t}{4}e^r)}, \quad \varphi(w^{(c)}) = -\frac{2}{3}w^{(c)3} - 2w^{(c)}, \quad \varphi''(w^{(c)}) = 2[w^{(c)} + \frac{1}{w^{(c)}}]. \tag{191}$$

For the saddle points to belong to the contour $C_w$, one needs $-\frac{t}{4}e^r \in [-\frac{1}{e}, 0]$, and up to redefinition of $r$ by a shift of $\frac{t}{4}$, we have the constraint $r \in\ ]-\infty, -1]$. This accounts to a redefinition of the initial condition $H \to H + \log(\frac{t}{4})$. For $w^2 \in [-1, 0]$, we choose the branch $W_0$ and for $w^2 \in\ ]-\infty, -1]$, we choose the branch $W_{-1}$. In the overall we have 16 purely imaginary saddle-point combinations, four for each variable which we denote by an index $i \in [1, 4]$. A saddle-point expansion yields

$$K_{12}(rt^{-\frac{1}{3}}, r't^{-\frac{1}{3}}) \simeq \frac{t^{1/3}}{16\pi i^2} \sum_{i,j=1}^{4} \frac{v_i^{(c)} - w_j^{(c)}}{v_i^{(c)} + w_j^{(c)}} \frac{1}{\sqrt{v_i^{(c)}}} \frac{1}{\sqrt{-w_j^{(c)}}} \sqrt{\frac{-v_i^{(c)}}{1 + v_i^{(c)2}}} \sqrt{\frac{-w_j^{(c)}}{1 + w_j^{(c)2}}}$$

$$\exp\left(\frac{1}{\sqrt{t}}[\varphi_r(v_i^{(c)}) + \varphi_{r'}(w_j^{(c)})]\right). \tag{192}$$

The square roots are not simplified as there exist different branches in the complex plane. At the very end we are interested in the $r = r'$ limit, hence we write $r' = r + \kappa$ and aim at taking the $\kappa = 0$. As $\kappa$ is small, we expand the critical point and the value of the rate function at the critical point

$$w_j^{(c)} = v_j^{(c)} + \frac{1}{2} \frac{v_j^{(c)}\kappa}{1 + v_j^{(c)2}}, \qquad \varphi_{r'}(w_j^{(c)}) = \varphi_r(v_j^{(c)}) - v_j^{(c)}\kappa. \tag{193}$$

Within the linear regime, the off-diagonal kernel reads

$$K_{12}(\frac{r}{t^{\frac{1}{3}}}, \frac{r+\kappa}{t^{\frac{1}{3}}}) \simeq \frac{t^{1/3}}{16\pi i^2} \sum_{i,j=1}^{4} \frac{v_i^{(c)} - v_j^{(c)} - \frac{1}{2}\frac{v_j^{(c)}\kappa}{1+v_j^{(c)2}}}{v_i^{(c)} + v_j^{(c)} + \frac{1}{2}\frac{v_j^{(c)}\kappa}{1+v_j^{(c)2}}} \frac{1}{\sqrt{v_i^{(c)}}} \frac{1}{\sqrt{-v_j^{(c)}}} \sqrt{\frac{-v_i^{(c)}}{1+v_i^{(c)2}}} \sqrt{\frac{-v_j^{(c)}}{1+v_j^{(c)2}}}$$

$$\exp\left(\frac{1}{\sqrt{t}}[\varphi_r(v_i^{(c)}) + \varphi_r(v_j^{(c)}) - v_j^{(c)}\kappa]\right). \tag{194}$$

From there, either we have three different combinations $W_0 \times W_{-1}$, $W_0 \times W_0$ and $W_{-1} \times W_{-1}$. The leading term of this expansion is obtained for $v_i^{(c)} = -v_j^{(c)}$, i.e. the saddle points are of the same Lambert branch but with opposite sign. This cancels the $\varphi$ functions in the exponential and the denominator in the sum is of order $\kappa$. We additionally rescale $\kappa$ by a factor $\sqrt{t}$ to obtain

$$K_{12}(\frac{r}{t^{\frac{1}{3}}}, \frac{r+\kappa\sqrt{t}}{t^{\frac{1}{3}}}) \simeq \sum_{i=1}^{2} \frac{1 + v_i^{(c)2}}{\kappa 4\pi t^{1/6}} \frac{1}{\sqrt{v_i^{(c)}}} \frac{1}{\sqrt{v_i^{(c)}}} \sqrt{\frac{-v_i^{(c)}}{1+v_i^{(c)2}}} \sqrt{\frac{v_i^{(c)}}{1+v_i^{(c)2}}} \exp\left(v_i^{(c)}\kappa\right). \tag{195}$$

Noticing that $1 + v_j^{(c)2}$ is positive for the branch $W_0$ and negative for the branch $W_{-1}$ and that whenever we take opposite saddle points we cross the branch cut of the square root, the off-diagonal kernel $K_{12}$ simplifies into the difference of two sine kernels

$$K_{12}(\frac{r}{t^{\frac{1}{3}}}, \frac{r+\kappa\sqrt{t}}{t^{\frac{1}{3}}}) \simeq \frac{1}{2\pi\kappa t^{1/6}} \left(\sin(\kappa\sqrt{-W_{-1}(-e^r)}) - \sin(\kappa\sqrt{-W_0(-e^r)})\right). \tag{196}$$

Taking $\kappa = 0$ yields the following density which vanishes at $r = -1$ and is strictly positive

$$\rho(rt^{-1/3}) \simeq \frac{1}{2\pi t^{1/6}} \left(\sqrt{-W_{-1}(-e^r)} - \sqrt{-W_0(-e^r)}\right)\theta(r \leq -1). \tag{197}$$

# D   Half-space propagator for the heat equation

We derive here the propagator of the heat equation in half-space(126). To do so, we shall find its fundamental solution for a point source, and as the heat equation is invariant by time-translation, we consider the problem

$$\partial_t Z(x,t) = \partial_x^2 Z(x,t), \tag{198}$$

along with a delta IC $Z(x, t = 0) = \delta(x - \varepsilon)$, $\varepsilon > 0$ and Robin b.c. $\partial_x Z(x,t)\mid_{x=0} = AZ(0,t)$.

We introduce the functions $v = \partial_x Z - AZ$, $\phi(x) = \delta'(x - \varepsilon) - A\delta(x - \varepsilon)$ and the heat kernel $G(x,t) = \frac{1}{\sqrt{4\pi t}}\exp(-\frac{x^2}{4t})\theta(t)$ along with $G(x,0) = \delta(x)$. As $v$ verifies the heat equation with Dirichlet b.c., i.e. $v(x = 0, t) = 0$, with $\phi$ as IC, we obtain its expression by the image method, i.e. the anti-symmetrization of the full-space kernel

$$
\begin{aligned}
v(x,t) &= \int_0^{+\infty} dy \left( G(x - y, t) - G(x + y, t) \right) \phi(y) \\
&= \partial_x G(x - \varepsilon, t) + \partial_x G(x + \varepsilon, t) - A\big( G(x - \varepsilon, t) - G(x + \varepsilon, t)\big).
\end{aligned}
\tag{199}
$$

We solve the ODE $v = \partial_x Z - AZ$ to obtain the partition function. For this, we need to introduce two constants $a$ and $C$ so that

$$
\begin{aligned}
Z(x,t) &= \int_a^x dy\, e^{A(x-y)} v(y,t) + Ce^{Ax} \\
&= \left[ (G(y - \varepsilon, t) + G(y + \varepsilon, t))e^{A(x-y)} \right]_a^x + 2A\int_a^x dy\, e^{A(x-y)} G(y + \varepsilon, t) + Ce^{Ax}.
\end{aligned}
\tag{200}
$$

The determination of $a$ and $C$ should enforce the matching with the IC of $Z$. Indeed, at $t = 0$, choosing $a = +\infty$ and $C = 0$ one obtains

$$Z(x,0) = \delta(x - \varepsilon) + \delta(x + \varepsilon) + 2Ae^{Ax}\int_{+\infty}^x dy\, e^{-Ay}\delta(y + \varepsilon). \tag{201}$$

Two of the three $\delta$ functions, $\delta(x + \varepsilon)$ and $\delta(y + \varepsilon)$, are zero as we define the partition function over the positive real line and as their support is on the negative real line. Therefore, the fundamental solution of the heat equation with Robin b.c. is

$$Z(x,t) = G(x - \varepsilon, t) + G(x + \varepsilon, t) - 2A\int_0^{+\infty} dy\, e^{-Ay} G(y + x + \varepsilon, t). \tag{202}$$

*Remark* D.1. If $A = 0$, then we find back the solution for the Neumann b.c. $\partial_x Z(x,t)\mid_{x=0} = 0$ which is $Z(x,t) = G(x - \varepsilon, t) + G(x + \varepsilon, t)$

*Remark* D.2. The infinite $A$ limit, i.e. the Dirichlet b.c. $Z(0,t) = 0$, and its first $1/A$ corrections are obtained by solving the integral in (202) exactly and expanding the result

$$Z(x,t) = G(x - \varepsilon, t) + G(x + \varepsilon, t) - Ae^{A(At+x+\varepsilon)}\mathrm{Erfc}\left( \frac{2At + x + \varepsilon}{2\sqrt{t}} \right). \tag{203}$$

The expansion for large $A$ gives

$$Z(x,t) = G(x - \varepsilon, t) - (1 - \frac{x + \varepsilon}{At} + \frac{x^2 + 2x\epsilon + \epsilon^2 - 2t}{2A^2 t^2} + \mathcal{O}(\frac{1}{A^3}))G(x + \varepsilon, t). \tag{204}$$

We finally interpret the solution as the propagator from a source point situated at $\varepsilon$ at time $\tau = 0$ going to the point $x$ in a time $t$, hence we define the propagator from $(y, t')$ to $(x, t)$, using the time-translation invariance.

$$\mathcal{G}(y, x, t', t) = G(x - y, t - t') + G(x + y, t - t') - 2A \int_0^{+\infty} dz \, e^{-Az} G(x + y + z, t - t'), \quad (205)$$

which is the propagator announced in (126).

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
