# Peer review of "Large fluctuations of the KPZ equation in a half-space"

_SciPost Physics, doi:SciPost Phys. 5, 032 (2018)_

## Round 1 · Referee Report · Anonymous · 2018-7-5

Strengths

1. Provides new exact formula for a few half-space cases.

2. Systematic approach for studying short time large deviation.

Weaknesses

1. Validity of the approximation is not clearly discussed.

2. There are repetitions of the same formulas without clear pointers.

3. Looks a bit like a collection of calculations.

Report

In this paper the authors study (mainly short time) large deviation properties of the KPZ equation. They use an explicit representation of a generating function of the height in the form of Fredholm determinant or Fredholm Pfaffian and applies the cumulant approximation in [33]. They show wide applicability of the method by studying a few cases of the KPZ equation in half-space. They also provide a new Fredholm Pfaffian formula for the half-space KPZ equation with the droplet and stationary situation.

The motivation of the study is sound. The new exact formulas for the half-space case are new and would be useful for future studies. On the other hand, their main results about the short time large deviation is based on what they call the cumulant approximation. A problem is that it is not clear how reliable this approximation is.
They write that in [33] it was observed that such an approximation is valid for a certain cases in full-space. There seems no guarantee that the same approximation is valid for other cases, but they do not seem to give serious discussions about the applicability of this approximation. The authors should provide clear and convincing arguments of the validity of the approximation or at least give some numerical evidence that the approximation seems to hold.

The presentation of the results are not optimal. They first present the main results in section 2. The authors should provide clearer pointers both in section 2 and in main texts. For example for the formula (5)(6), it is written that “These results are shown in Section 5”. Subsection 5.1 should be more appropriate. In addition, in subsection 5.1, the same formulas appear as (63)(64) without any notice. This should be pointed out clearly. In fact one may omit (63)(64) and refer to (5)(6). Similarly, there are some repetitions of the contents of section 2.2 and the main texts. For example (13) and (43) are the same. The connection should be clearly stated.

The paper will be reconsidered after a revision.

Requested changes

1. Give clear and convincing arguments for the validity of their approximation.

2. Give clear pointers to the repeated formulas.

---

## Round 2 · Referee Report · Anonymous · 2018-9-10

Report

In the revision, they take into the comments appropriately and provide an information about the validity of the cumulant expansion (basically they will be given in [45] in future) and also give cross-references among various parts of the paper. As the reviewer wrote in the original review, the paper contains enough interesting results. He now recommends a publication of this article.

---

## Round 2 · Author Response

Dear Editor and Referee,

We are grateful for the efforts in reviewing our manuscript. We thank the referee for his constructive comments of our paper. In the following, we believe we answer the concerns raised by the referee and list all changes we made in the resubmitted version.

Sincerely yours,

Alexandre Krajenbrink and Pierre Le Doussal
* * *
Response to the Referee 
* * *
The referee raised two main concerns.

1. The referee asked that we discuss more clearly the validity of the cumulant approximation. This is now elaborated in the main text Section 3.1 below Eq. 19. 

We argue that the cumulant approximation can be explained by a law of large number for the set of points generated by the Pfaffian point process in the short-time regime. The observable we are interested in (the sum of functions $\phi$ in Eq. 17) gets self-averaged as most points of the process are involved in the sum. This explains why the first cumulants dominates higher order ones. 

We would like to mention that these arguments are fully confirmed by explicit calculations which show that the cumulant expansion can be made systematic, and this, together with numerical evidence, will be presented in work in preparation, as we have now indicated
in the text - Ref  [45].

We would finally like to stress that the goal of our manuscript is to apply the cumulant approximation previously used in full-space cases of the KPZ equation to numerous half-space cases where the Pfaffian representation of the generating function is available. This is particularly helpful to unravel universal properties of the large deviations of the solution of the KPZ equation at short time. We hope this will help to get a broader picture of the behavior of the general solution for which one does not necessarily have a determinant or Pfaffian representation.

2. The other concern the referee raised is about clear pointing and cross-referencing in our manuscript. We thank the referee for this remark and added pointers to all our formulas to ensure that all derivations are made extremely clear for the reader.

---

## Round 2 · List of Changes

List of changes :

- We added in Section 3.1 below Eq. 19 a discussion about the validity of the cumulant approximation.  

- We added numerous cross-references and pointers all along the manuscript to ensure that all derivations are made extremely clear for the reader.

- We omitted the repetition of the expression of the new kernel for the hard-wall case by pointing in Section 5.1 to the result announced in Section 2.1 as advised by the referee.

You are currently on this page

Resubmission 1804.08800v2 on 27 July 2018

---

## Editorial Decision

published